# Passive epidemiological surveillance in wildlife in Costa Rica identifies pathogens of zoonotic and conservation importance

Fernando Aguilar-Vargas[1,2☯], Tamara Solorzano-Scott[1☯], Mario Baldi[3], Elías Barquero-Calvo[3,4], Ana Jiménez-Rocha[5], Carlos Jiménez[3,4,5,6], Marta Piche-Ovares[6], Gaby Dolz[7], Bernal León[2], Eugenia Corrales-Aguilar[8], Mario Santoro[9], Alejandro Alfaro-Alarcón[1]*

1 Departamento de Patología, Escuela de Medicina Veterinaria, Universidad Nacional, Costa Rica, 2 Servicio Nacional de Salud Animal, Ministerio de Agricultura y Ganadería, Costa Rica, 3 Programa de Investigación en Enfermedades Tropicales, Escuela de Medicina Veterinaria, Universidad Nacional, Costa Rica, 4 Laboratorio de Bacteriología, Escuela de Medicina Veterinaria, Universidad Nacional, Costa Rica, 5 Laboratorio de Parasitología, Escuela de Medicina Veterinaria, Universidad Nacional, Costa Rica, 6 Laboratorio de Virología, Escuela de Medicina Veterinaria, Universidad Nacional, Costa Rica, 7 Laboratorio de Zoonosis y Entomología, Escuela de Medicina Veterinaria, Universidad Nacional, Costa Rica, 8 Faculty of Microbiology, Virology-CIET (Research Center for Tropical Diseases), Universidad de Costa Rica, Costa Rica, 9 Department of Integrative Marine Ecology, Stazione Zoologica Anton Dohrn, Italy

☯ These authors contributed equally to this work.
* alejandro.alfaro.alarcon@una.cr

## Abstract

Epidemiological surveillance systems for pathogens in wild species have been proposed as a preventive measure for epidemic events. These systems can minimize the detrimental effects of an outbreak, but most importantly, passive surveillance systems are the best adapted to countries with limited resources. Therefore, this research aimed to evaluate the technical and infrastructural feasibility of establishing this type of scheme in Costa Rica by implementing a pilot program targeting the detection of pathogens of zoonotic and conservation importance in wildlife. Between 2018 and 2020, 85 carcasses of free-ranging vertebrates were admitted for post-mortem and microbiology analysis. However, we encountered obstacles mainly related to the initial identification of cases and limited local logistics capacity. Nevertheless, this epidemiological surveillance scheme allowed us to estimate the general state of health of the country's wildlife by establishing the causes of death according to pathological findings. For instance, 60% (51/85) of the deaths were not directly associated with an infectious agent. Though in 37.6% (32/85) of these cases an infectious agent associated or not with disease was detected. In 27.1% (23/85) of the cases, death was directly related to infectious agents. Furthermore, 12.9% (11/85), the cause of death was not determined. Likewise, this wildlife health monitoring program allowed the detection of relevant pathogens such as Canine Distemper Virus, *Klebsiella pneumoniae*, *Angiostrongylus* spp., *Baylisascaris* spp., among others. Our research demonstrated that this passive surveillance scheme is cost-effective and feasible in countries with limited resources. This passive surveillance can be adapted to the infrastructure dedicated to monitoring diseases in productive animals according to the scope and objectives of monitoring

**Data Availability Statement:** All relevant data are within the paper and its Supporting Information files.

**Funding:** The authors received no specific funding for this work.

**Competing interests:** The authors have declared that no competing interest exist.

wildlife specific to each region. The information generated from the experience of the initial establishment of a WHMP is critical to meeting the challenges involved in developing this type of scheme in regions with limited resources and established as hotspots for emerging infectious diseases.

## Introduction

Zoonotic diseases directly threaten public health systems, generating costs in medical treatment, outbreak control, and overloading health systems. In addition, it generates significant losses due to the slaughter of livestock and the affectation of other domestic animals [1,2]. Examples of how these diseases can impact public health, animal health, and wildlife have been the recent outbreaks of yellow fever and West Nile virus, which show the need to have the infrastructure and diagnostic capacity to ensure constant surveillance of potentially zoonotic agents [3,4].

Wildlife populations act as reservoirs and can play various roles in the epidemiology of numerous pathogens [5–7]. These roles assign to wildlife the important function of sentinels of the health of ecosystems and allow early detection of environmental alterations and the distribution, re-emergence, or emergence of certain pathogens in a specific region [8,9].

Tropical regions are among the areas of most extraordinary natural diversity with a concomitant high diversity of pathogens and, thus, a high potential for disease emergence [10,11]. Moreover, this risk has increased drastically because of anthropogenic pressures linked to over-exploitation of natural resources and increased land use change, increasing the possibility of contact between wildlife, domestic animals, and humans [12,13].

One of the preventive strategies against the risk of epidemic events promoted by the World Organization for Animal Health (OIE) and the World Health Organization (WHO) is to increase the efforts to establish early detection mechanisms for pathogens, of both zoonotic and conservation importance, via Wildlife Health Monitoring Programs (WHMP) [14–16].

One of the first steps to knowing the health status of the wildlife in a region is monitoring through passive surveillance, which identifies the causes of mortality in a range of species based on their pathological profiles through post-mortem examinations. This approach offers advantages like cost-effectiveness and the ability to carry out convenience samplings, taking advantage of the established infrastructure and diagnostic capacity. Furthermore, when these schemes are set in the long term, it has been shown that they provide the core information for decision-making and the establishment of policies, norms, and strategies, prioritizing disease prevention, even when the sampling is biased and with incomplete geographic coverage [17–20].

In Latin America have been made some significant efforts to improve epidemiological surveillance systems aimed at animal health. Some national programs are installed and functioning perfectly where wild animals are used as sentinels to monitor specific diseases [21,22]. However, there are still no monitoring programs for the general health status of wildlife, making clear the need to optimize and expand the coverage of these schemes [23,24]. For example, according to the U.S. Department of Agriculture, Costa Rica has the infrastructure and maintains adequate surveillance programs to detect and control zoonotic diseases in livestock [25]. However, it does not contemplate local wildlife within its scheme as it should [26].

Several pathogens, such as zoonotic parasites, vector-borne infectious agents, and direct transmission viruses, have been identified in Costa Rican wildlife [27–38]. This evidence reflects the urgency of establishing a permanent WHMP, where aspects such as general health

status and monitoring of zoonotic pathogens in wildlife are considered, facilitating knowledge of the ecoepidemiology of these agents at the local level.

Countries with limited resources, such as Costa Rica, face severe financial and logistical restrictions in monitoring the health and circulation of pathogens in wildlife. Nevertheless, in the short term must extend the coverage of this type of program to tropical regions. Therefore, this research aims to evaluate the technical and infrastructural feasibility of establishing this type of scheme in Costa Rica by implementing a pilot program for passive epidemiological surveillance of wildlife. Although we encountered obstacles such as a lack of data collection legislation and a willingness to cooperate among agencies, our research demonstrated the logistical capacity and that it is possible to adapt the established infrastructure to implement this program. Furthermore, this allowed wild animal carcasses to be analyzed, detecting zoonotic pathogens and pathogens of conservation importance.

## Material and methods

### Statement of ethics

All samples were obtained from dead wildlife (found dead in the field or euthanized after veterinary care in specialized centers). The study was approved by the Ministry of Environment and Energy (MINAE) (wildlife authority) through permits (R-SINAC-PNI: -ACAT-040, ACAHN-18, ACTo-022, ACT-OR-DR-43, ACG-026, ACLAC-039, ACLAP-023, ACOPAC-005, ACC-037), and with the support of the animal health authority, the National Animal Health Service through the office (SENASA-DG-0277-18).

### WHMP schema proposal and case definition

For the implementation of a WHMP, a passive epidemiological surveillance scheme was proposed adapting the current country's technical diagnostic resources and infrastructure. To create a network for detecting dead and diseased wild animals, officials from the wildlife management centers and officials from wildlife authorities reported cases and voluntarily sent specimens. Officials were encouraged to send complete carcasses from free-ranging vertebrates after death due to any associated disease or trauma, both found dead in the field or deceased in management centers. Carcasses of animals that remained more than 48 hours in the management centers before death, received medication, or were frozen for more than a week were excluded from the study. The proposed WHMP scheme is shown in Fig 1.

Basic information was requested and registered for every sample submission: geographic location, the standard and scientific name of the animal, clinical signs, and any information considered relevant to the case, following the scheme recommended by the OIE for the notification of cases for disease surveillance system in wild animals [16,39]. All carcasses were shipped under refrigerated conditions at 2–8°C.

### Pathological analysis

The carcasses received were classified by autolysis degree according to an established scale of one to five [40]. Thus, ranging from a fresh carcass or recently dead animal (grade 1) to advanced decomposition (grade 4) and partial, mummified carcasses or skeletal remains (grade 5). Only carcasses with grades 1 to 3 were included in the study for post-mortem analysis and tissue sampling [41]. Therefore, 96 specimens were received, of which 85 were admitted to the study. Specimens were divided by sex and age according to the development of sexual organs and phenotypic characteristics of the species. Also, they were divided by taxonomic order.

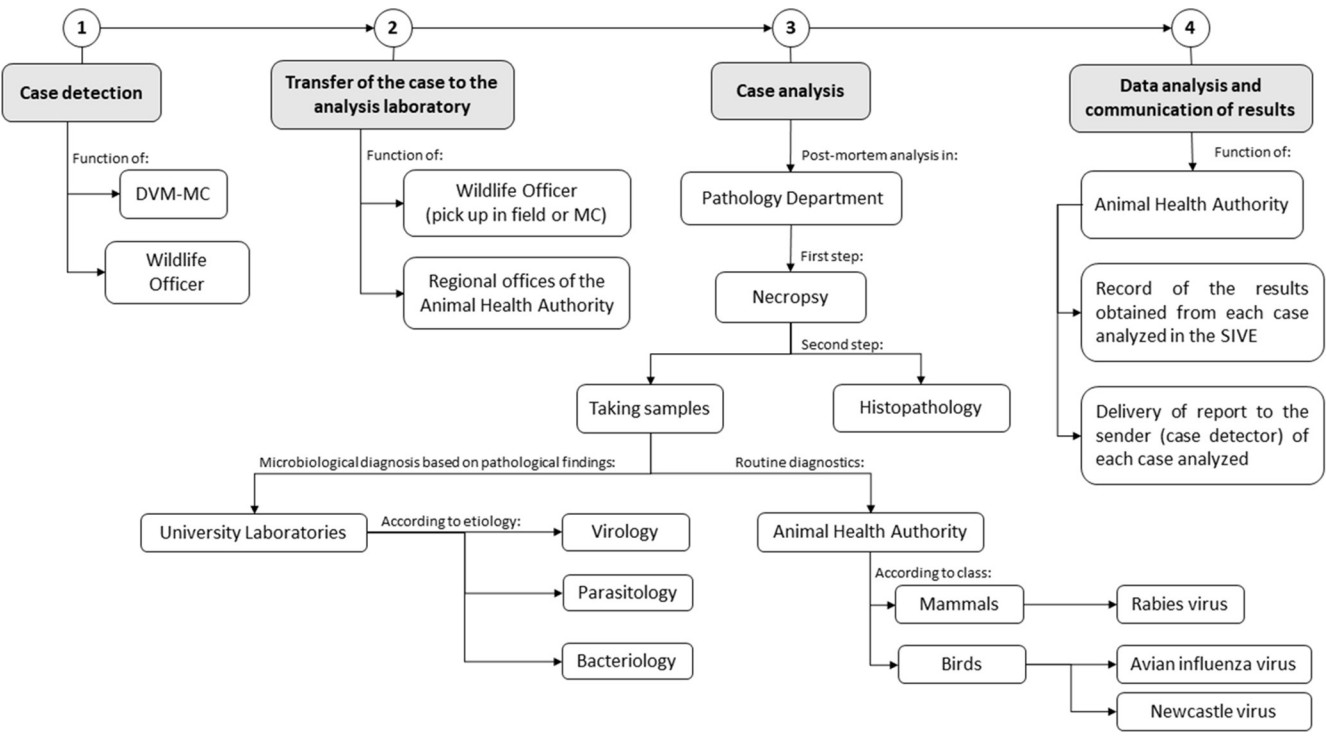

**Fig 1. Pilot WHMP work scheme design proposal.** DVM-MC: Doctor of veterinary medicine of wildlife management centers; MC: Wildlife management centers; PD: Pathology Department of Escuela de Medicina Veterinaria, Universidad Nacional.

All morphological findings were recorded. In addition, tissue samples were taken for routine histopathological and microbiological analysis as required. Tissue samples for histopathology were processed based on standard routing protocols [41].

## Detection of different infectious agents

**Virus detection.** Molecular methods were used to detect different viral agents. All molecular methods were done in the presence of positive and negative controls. The samples analyzed were fresh tissues collected sterile during post-mortem analysis. In addition, we performed RNA extraction using the commercial kit DNeasy Blood and Tissue (QIAGEN, Venlo, The Netherlands), following the manufacturer's recommendations. The methods used and the samples collected are specified in Table 1.

**Detection of protozoan parasites'.** Confirmation was performed using molecular techniques for pathogen identification when a previous presumptive protozoa presence was established in the histopathological study. All molecular methods were done in the presence of positive and negative controls. Tissue samples previously embedded in paraffin were used for this purpose. The deparaffinization procedure was done using xylol washes following the method recommended to perform DNA extraction from the tissue [57]. We performed DNA extraction using the commercial kit DNeasy Blood and Tissue (QIAGEN, Venlo, The Netherlands) according to the manufacturer's instructions. The methods used and the samples collected are specified in Table 1.

**Bacteriological detection.** Tissue samples from animals with inflammatory processes (suppurative or abscesses) were cultured following standard bacteriological procedures. For bacterial isolation, samples were inoculated on non-selective and selective agar media.

**Table 1. Molecular techniques for the detection of viral agents and protozoa.**

| Infectious agent | Target region | Method | Primer | Sequence | Reference protocol | Used material |
|---|---|---|---|---|---|---|
| **Canine Distemper Virus (CDV).** | N gene | Nested RT-PCR | First round: CDV-1F | 5'- ACT GCT CCT GAT ACT GC-3' | Da Budaszewski et al., 2014. [42] | Tissue[a] |
| | | | CDV-2R | 5'- TTC AAC ACC RAC YCC C-3' | | |
| | | | Second round: CDV-3F | 5'- ACA GRA TTG CYG AGG ACY TRT-3' | | |
| | | | CDV-4R | 5'- CAR RAT AAC CAT GTA YGG TGC-3' | | |
| **Alphaviruses.** | nsP4 | Nested RT-PCR | First round: – | 5'- TTT AAG TTT GGT GCG ATG ATG AAG TC-3' (500 nM) | Grywna et al., 2010. [43] | Tissue[a] |
| | | | | 5'- GCA TCT ATG ATA TTG ACT TCC ATG TT-3' (500 nM) | | |
| | | | Second round: – | 5'-GGT GCG ATG ATG AAG TCT GGG ATG T-3' (200nM) | | |
| | | | | 5'- CTA TGA TAT TGA CTT CCA TGT TCA TCC A-3' (100 nM) | | |
| | | | | 5'-CTA TGA TAT TGA CTT CCA TGT TCA GCC A-3' (100 nM) | | |
| **Flaviviruses.** | NS5 gene | Semi-nested RT-PCR | First round: MAMD | 5'- AAC ATG ATG GGR AAR AGR GAR AA-3' | Scaramozzino et al., 2001. [44] | Tissue[a] |
| | | | cFD2 | 5'-GTG TCC CAG CCG GCG GTG TCA TCA GC-3' | | |
| | | | Second round: FS 778 | 5'-AAR GGH AGY MCD GCH ATH TGG T-3' | | |
| | | | cFD2 | 5'-GTG TCC CAG CCG GCG GTG TCA TCA GC-3' | | |
| **Avian Influenza virus (AI).** | matrix (M) gene | qRT-PCR | M + 25 | 5'-AGA TGA GTC TTC TAA CCG AGG TCG-3' | Spackman et al., 2002. [45] | Tissue and swab [b] |
| | | | M 124 | 5'-TGC AAA AAC ATC TTC TTC AAG TCT CTG-3' | | |
| | | | M + 64 | 5'-FAM–TCA GGC CCC CTC AAA GCC GA–TAMRA-3' | | |
| **Rabies virus.** | Nucleoprotein | RT–PCR | RAB504 | 5'-TAT ACT CGA ATC ATG AAT GGA GGT CGA CT-3' | Primers: Oliveira et al. 2010. [46] Protocol: Carnieli et al. 2008 [47] | Tissue[c] |
| | | | RAB304 | 5'-ACG CTT AAC AAC AAR ATC ARA G-3' | | |
| **Newcastle virus.** | Fusion gene, F0 | RT-PCR | NCD3 | 5'-GTC AAC ATA TAC ACC TCA TC-3' | STAUBER, 1995. [48] | Tissue and swab [b] |
| | | | NCD4 | 5'-GGA GGA TGT TGG CAG CAT T-3' | | |
| ***Toxoplasma gondii.*** | 529bp repetitive segment | PCR | Tox-8 | 5'-CCC AGC TGC GTC TGT CGG GAT-3' | Homan et al., 2000. [49] Reischl et al., 2003. [50] | FFPE[d] |
| | | | Tox-11 | 5'-GCG TCG TCT CGT CTA GAT CG-3' | | |
| ***Trypanosoma cruzi.*** | 18S rRNA gene | Nested PCR | First round: SSU4_F | 5'-GTG CCA GCA CCC GCG GTA AT-3' | First round primer: Pinto et al., 2015. [51] Second round primer: Noyes et al., 1999. [52] Protocol: Aleman et al., 2017. [53] Murphy & O'Brien, 2007.[54] | FFPE[e] |
| | | | 18Sq1R | 5'-CCA CCG ACC AAA AGC GGC CA-3' | | |
| | | | Second round: SSU561F | 5'-TGG GAT AAC AAA GGA GCA-3' | | |
| | | | SSU561R | 5'-CTG AGA CTG TAA CCT CAA AGC-3' | | |

(*Continued*)

**Table 1.** (Continued)

| Infectious agent | Target region | Method | Primer | Sequence | Reference protocol | Used material |
|---|---|---|---|---|---|---|
| *Leishmania* spp. | Kinetoplast | PCR | 13A | 5'- GTG GGG GAG GGG CGT TCT-3' | Medeiros et al. 2002. [55] Sosa-Ochoa et al. 2015. [56] | FFPE[f] |
| | | | 13B | 5'-ATT TTA CAC CAA CCC CCA GTT-3' | | |

FFPE: Formalin-fixed paraffin-embedded.

[a] brain and lung.

[b] Lung and Trachea tissue and cloacal swab.

[c] hippocampus, cerebellum, and medulla oblongata.

[d] spleen, lung, and liver.

[e] heart.

[f] spleen.

Significant bacterial growth was identified using the automated VITEK-2 Compact system, software version 8.02 (bioMérieux, Marcy l'Etoile, France). VITEK test cards for Gram-negative [GN], Gram-positive [GP], and anaerobes [ANC] were used for identification according to the manufacturer's instructions.

**Identification of metazoan parasites.** All the parasites in the carcasses were collected and washed with physiological saline, preserved in alcohol, acetic acid, and formalin (AFA) solution. No more than one week after collection, they underwent identification to the genus level through morphometric characteristics [58]. Physical and morphometric characteristics were recognized after fixation and clarification with Hoyer's solution by light microscopy [59–61]. In addition, processed cestodes were stained with dilute Harris' hematoxylin solution.

## Information management, geocoding, and spatial analysis

The information on each case was included in the epidemiological surveillance information system (SIVE) from the animal health authority. Each case was geocoded using the latitude and longitude generated by GPS of the point where the specimen was found by field personnel. When the GPS was unavailable, they were geocoded using the latitude and longitude of the approximate location where they were found, and this was generated by Google Earth Pro v7.3 (2021, Google Inc.). With the georeferenced points of each sample admitted created a map using ArcGIS 10.7 (ERSI), according to territorial division by conservation area: Arenal Huetar Norte Conservation Area (ACAHN); Arenal Tempisque Conservation Area (ACT); Central Conservation Area (ACC); Guanacaste Conservation Area (ACG); La Amistad Caribe Conservation Area (ACLAC); La Amistad Pacífico Conservation Area (ACLAP); Osa Conservation Area (ACOSA); Pacífico Central Conservation Area (ACOPAC); Tempisque Conservation Area (ACT); Tortuguero Conservation Area (ACTo). Additionally, a feedback report was sent to the field staff with the relevant findings per case.

## Results

### Participation in the WHMP and distribution of cases by age, sex, and taxonomic classification

The notification of cases was made by officials from the wildlife authority, with 24.7% (21/85) of the cases and 75.3% (64/85) by officials from wildlife management centers. Only four management centers reported and sent cases for analysis. The conservation areas with the most

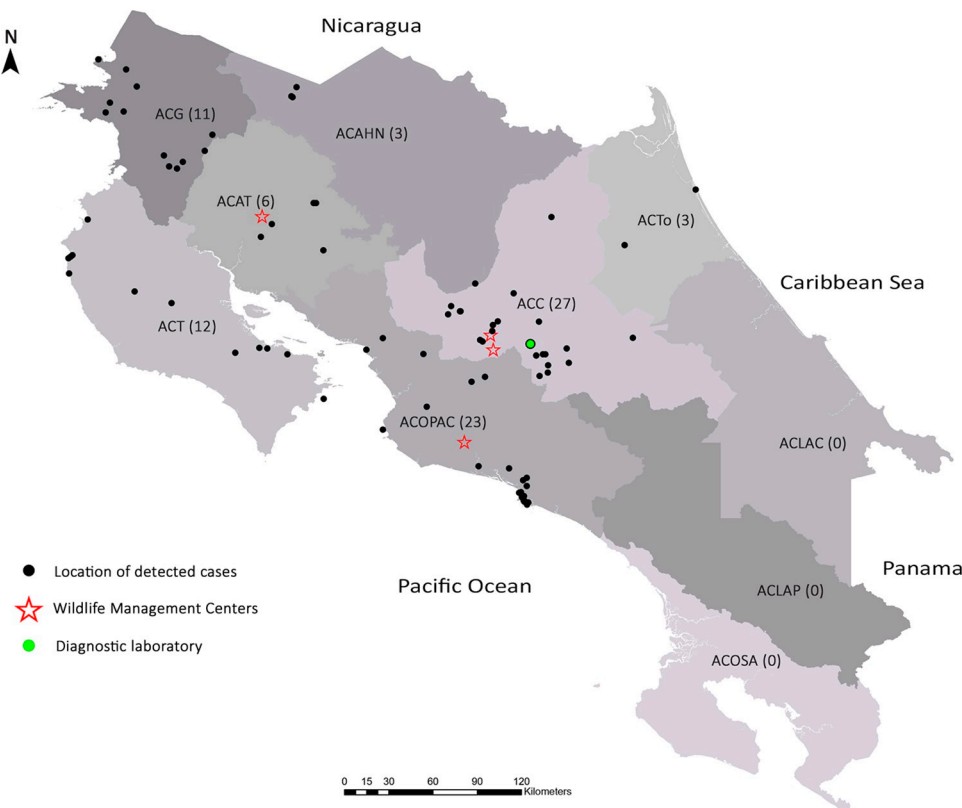

**Fig 2. Geocoding of the cases analyzed by conservation area.** The number corresponds to the cases analyzed in each conservation area. Wildlife management centers shown are those that collaborated with the WHMP.

significant participation in the WHMP were the same ones where the participating wildlife management centers were located. The geographical location of the management centers, diagnostic laboratory, and cases analyzed is shown in Fig 2. The conservation areas where there was no participation are those located furthest from the diagnostic laboratory and with significant obstacles for shipment, as mentioned in Table 2.

Of the 85 specimens admitted to the study, there was an age distribution of 27.1% (23/85) young animals and 72.9% (62/85) adults. The sex distribution was 56.5% (48/85) males and 43.5% (37/85) females. According to the taxonomic order, we received 29.4% (25/85) Carnivora, 29.4% (25/85) Primate, 12.9% (11/85) Pilosa, 5.9% (5/85) Didelphimorphia, 4.7% (4/85) Rodentia, 4.7% (4/85) Artiodactyla, 2.3% (2/85) Cingulate, 2.3% (2/85) Pelecaniformes, 2.3% (2/85) Accipitriformes, 2.3% (2/85) Anseriformes, 1.2% (1/85) Ciconiiformes, 1.2% (1/85) Piciformes and 1.2% (1/85) Coraciiformes. The geographical distribution of admitted cases by conservation area is shown in Fig 2.

## Identification of causes of death according to pathological findings

According to pathological findings, the distribution of the presumptive cause of death corresponded to 60% (51/85) of death not associated with an infectious agent. Of these, 54.1% (46/85) associated with traumatic events (mainly roadkill and electrocution), 2.4% (2/85) with a degenerative disease, and in 3.5% (3/85) of cases, death was presumptively associated with intoxication. Additionally, of individuals with a cause of traumatic death, 37.6% (32/85)

**Table 2. Participation in the WHMP of detectors of cases and obstacles found in each conservation area.**

| Conservation area | Number of cases | Cases detector | | Obstacles to sending cases |
|---|---|---|---|---|
| | | Wildlife Officer | MC | |
| ACAHN | 3 | 0 | 3 | Inability to store. Coordination problems with the health agency for the transport of specimens. Few rescue centers motivated to participate. |
| ACAT | 6 | 0 | 6 | Coordination problems with the wildlife agency to submit specimens. |
| ACC | 27 | 6 | 21 | No significant obstacles. |
| ACG | 11 | 7 | 4 | Coordination problems with the health agency for the transport of specimens. |
| ACLAC | 0 | 0 | 0 | Few rescue centers motivated to participate. Coordination problems with the wildlife agency to submit specimens. Coordination problems with the health agency for the transport of specimens. Distant from the diagnostic laboratory. |
| ACLAP | 0 | 0 | 0 | There are no rescue centers in the region. Coordination problems with the wildlife agency to submit specimens. |
| ACOSA | 0 | 0 | 0 | There are no rescue centers in the region. Distant from the diagnostic laboratory. |
| ACOPAC | 23 | 0 | 23 | Coordination problems with the wildlife agency to submit specimens. |
| ACT | 12 | 5 | 7 | Few rescue centers in the region. Inability to store |
| ACTo | 3 | 3 | 0 | There are no rescue centers in the region. Insufficient field staff. Coordination problems with the health agency for the transport of specimens. |

ACAHN: Conservation area Arenal Huetar Norte; ACT: Conservation area Arenal Tempisque; ACC: Conservation area Central; ACG: Conservation area Guanacaste; ACLAC: Conservation area La Amistad Caribe; ACLAP: Conservation area La Amistad Pacifico; ACOSA: Conservation area Osa; ACOPAC: Conservation area pacific central; ACT: Conservation area Tempisque; ACTo: Conservation area Tortuguero.

concomitantly presented some infectious agent with or without an associated disease (24 with gastrointestinal and pulmonary metazoan parasites, three with bacteria, one with protozoa, and four with multiple microorganisms). In 27.1% (23/85) of cases, death was directly related to infectious agents, ten presented lesions associated with viruses, five with metazoan parasites, two with protozoan parasites, one with bacteria, and five presented lesions associated with multiple etiologies. In 12.9% (11/85) of cases, the cause of death was not determined. The absolute and relative values of the causes of death for each taxonomic group according to the presence of infectious agents are specified in Table 3.

## Infectious agents detected in the WHMP

Ten viruses, seven protozoa, and seven bacteria were identified in mammalian specimens. In 22 cases, these pathogens were involved with lesions or systemic disease, of which 19 were directly associated with the cause of death of mammals. Only *Sarcocystis* spp. detected in two cases was an incidental finding. Additionally, 38 mammals had metazoan parasites. Multi-parasitosis was observed in 15.3% (13/85) of the cases. Parasites such as *Prosthenorchis* spp. (n = 15), *Angiostrongylus* spp. (n = 6), and *Cilycospirura* spp. (n = 1) were responsible for severe parasitosis with systemic disease. Some of the lesions, such as pyogranulomatous abscessing bronchopneumonia and nodular and sclerosing gastritis associated with infectious agents, are observed in Fig 3 (see legend). In 50.6% (43/85) of the cases, the mammals presented infectious agents with a zoonotic potential, such as *Klebsiella pneumoniae*, *Toxoplasma gondii*, *Angiostrongylus* spp. The etiological agents identified by taxonomic groups and the number of specimens analyzed are specified in Table 4.

All birds submitted were evaluated for virus presence (n = 9); two of these were positive for flaviviruses. Additionally, three birds had metazoan parasites. Most of the pathogens identified were directly associated with the cause of the death of birds. Only *Procyrnea* spp. identified in one case was an incidental finding. In 2.3% (2/85) of the cases, the birds presented infectious

**Table 3. Absolute and relative values of the causes of death for each taxonomic group.**

| Cause of Death / Taxon | DAIA | DNAIA-PD | DNAIA-IAD | DNAIA | UD |
|---|---|---|---|---|---|
| **Mammals** | | | | | |
| Carnivora | 40% (10/25) | 28% (7/25) | 16% (4/25) | 8% (2/25) | 8% (2/25) |
| Primate | 32% (8/25) | 36% (9/25) | 16% (4/25) | 8% (2/25) | 8% (2/25) |
| Pilosa | 0% (0/11) | 9.1% (1/11) | 27.3% (3/11) | 45.4% (5/11) | 18.2% (2/11) |
| Didelphimorphia | 20% (1/5) | 0% (0/5) | 60% (3/5) | 0% (0/5) | 20% (1/5) |
| Rodentia | 25% (1/4) | 0% (0/4) | 0% (0/4) | 25% (1/4) | 50% (2/4) |
| Artiodactyla | 25% (1/4) | 0% (0/4) | 0% (0/4) | 75% (3/4) | 0% (0/4) |
| Cingulate | 0% (0/2) | 0%% (0/2) | 0%% (0/2) | 100% (2/2) | 0%% (0/2) |
| **Birds** | | | | | |
| Pelecaniformes | 100% (2/2) | 0% (2/2) | 0% (2/2) | 0% (2/2) | 0% (2/2) |
| Accipitriformes | 0% (0/2) | 0% (0/2) | 50% (1/2) | 50% (1/2) | 0% (0/2) |
| Anseriformes | 0% (0/2) | 0% (0/2) | 0% (0/2) | 100% (2/2) | 0% (0/2) |
| Ciconiiformes | 0% (0/1) | 0% (0/1) | 0% (0/1) | 0% (0/1) | 100% (1/1) |
| Piciformes | 0% (0/1) | 0% (0/1) | 0% (0/1) | 100% (1/1) | 0% (0/1) |
| Coraciiformes | 0% (0/1) | 0% (0/1) | 0% (0/1) | 0% (0/1) | 100% (1/1) |
| **Total** | 27.1% (23/85) | 20% (17/85) | 17.6% (15/85) | 22.4% (19/85) | 12.9% (11/85) |

DAIA: Death associated with an infectious agent; DNAIA-PD: Death not associated with an infectious agent, with a pre-existing infectious disease; DNAIA-IAD: Death not associated with an infectious agent, with infectious agent detection; DNAIA: Death not associated with an infectious agent; UD: Undetermined death.

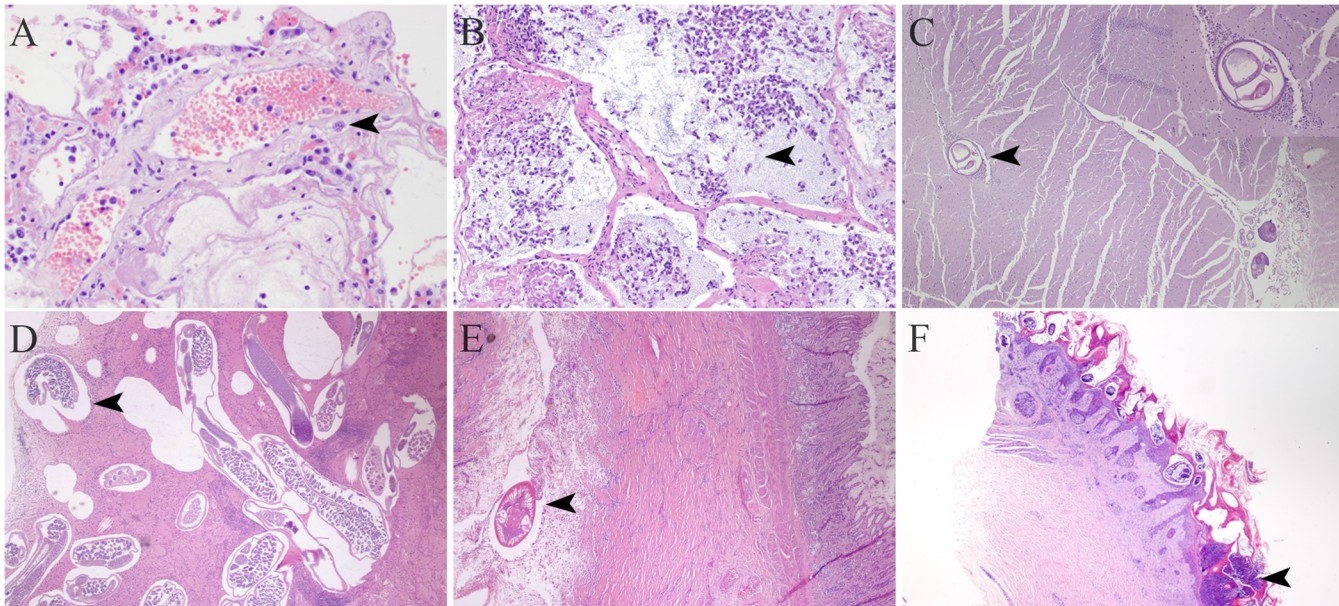

**Fig 3. Infectious agents in lesions identified in wild animals.** A) Lung (*Alouatta palliata*-howler monkey). Lymphoplasmacytic pneumonia with the presence of tissue cyst, morphology compatible with *Toxoplasma gondii*, confirmation by PCR (arrowhead; H&E 600x). B) Lung (*Alouatta palliata*-howler monkey). Pyogranulomatous abscessing bronchopneumonia with intralesional bacteria *Klebsiella pneumonia*, confirmation by culture (arrowhead; H&E 200x). C) Brain (*Didelphis marsupialis*-opossum). Presence of nematode *Angiostrongylus* spp. identified by morphology (arrowhead; H&E 400x). Inset: Nematode magnification (H&E 200x). D) Lung (*Cebus imitator*-white-faced monkey). Bronchopneumonia associated to multiple Nematodes, *Filariopsis* spp. identified by morphology (more cuts of the female in microphotograph) (arrowhead; H&E 40x). E) Stomach (*Herpailurus yagouaroundi*-jaguarundi). Nodular and sclerosing gastritis associated with multiple *Cylicospirura* spp. Nematodes identified by morphology (arrowhead; H&E 40x). F) Skin (*Sphiggurus mexicanus*-porcupine) Pyogranulomatous and eosinophilic dermatitis associated with massive infestation of *Sarcoptex* spp. (arrowhead; H&E 400x). Inset: Mites magnification (H&E 100x).

**Table 4. Number of infectious agents tested and positive in mammals according to etiology.**

| Mammalian taxonomic groups / infectious agent | | Primate | Carnivora | Pilosa | Didelphimorphia | Rodentia | Artiodactyla | Cingulate |
|---|---|---|---|---|---|---|---|---|
| Viral | CDV (n = 18) | 0 | 10 | 0 | 0 | 0 | 0 | 0 |
| | Alphaviruses (n = 9) | 0 | 0 | 0 | 0 | 0 | 0 | 0 |
| | Flaviviruses (n = 9) | 0 | 0 | 0 | 0 | 0 | 0 | 0 |
| | Influenza virus (n = 8) | 0 | 0 | 0 | 0 | 0 | 0 | 0 |
| | Rabies virus (n = 76) | 0 | 0 | 0 | 0 | 0 | 0 | 0 |
| Bacterial | *C. perfringens* (n = 18) | 0 | 0 | 0 | 0 | 0 | 1 | 0 |
| | *E. coli* (n = 18) | 1 | 0 | 0 | 0 | 0 | 0 | 0 |
| | *K. pneumoniae* (n = 18) | 1 | 0 | 0 | 0 | 0 | 0 | 0 |
| | *T. pyogenes.* (n = 18) | 0 | 0 | 0 | 0 | 1 | 0 | 0 |
| | *S. aureus* (n = 18) | 1 | 1 | 1 | 0 | 0 | 0 | 0 |
| | *Mycobacterium* spp. (n = 18) | 0 | 0 | 0 | 0 | 0 | 0 | 0 |
| Protozoan parasites | *Toxoplasma gondii* (n = 4) | 2 | 0 | 0 | 0 | 0 | 0 | 0 |
| | *Trypanosoma* spp. (n = 14) | 0 | 0 | 0 | 3 | 0 | 0 | 0 |
| | *Leishmania* spp. (n = 8) | 0 | 0 | 0 | 0 | 0 | 0 | 0 |
| | *Sarcocystis* spp. (n = 5) | 0 | 1 | 0 | 0 | 0 | 1 | 0 |
| **Metazoan parasites** [1] | *Angiostrongylus* spp. | 0 | 5 | 0 | 1 | 0 | 0 | 0 |
| | *Dirofilaria* spp. | 0 | 4 | 0 | 0 | 0 | 0 | 0 |
| | *Dipetalonema* spp. | 5 | 0 | 2 | 0 | 0 | 0 | 0 |
| | *Gnathostoma* spp. | 0 | 0 | 0 | 1 | 0 | 0 | 0 |
| | *Baylisascaris* spp. | 0 | 1 | 0 | 0 | 0 | 0 | 0 |
| | *Ancylostoma* spp. | 0 | 1 | 0 | 0 | 0 | 0 | 0 |
| | *Cylicospirura* spp. | 0 | 1 | 0 | 0 | 0 | 0 | 0 |
| | *Prosthenorchis* spp. | 10 | 5 | 0 | 0 | 0 | 0 | 0 |
| | *Macracanthorhynchus* spp. | 0 | 1 | 0 | 0 | 0 | 0 | 0 |
| | *Spirometra* spp. | 0 | 2 | 0 | 0 | 0 | 0 | 0 |

n: Number tested.

[1] only zoonotic metazoan parasites are shown.

agents with zoonotic potential, such as *Contracaecum* spp. The etiological agents identified in birds and the number of samples analyzed are specified in Table 5.

## Geospatial distribution of detected infectious agents and their accumulation by geographic region

We established the distribution of the most frequently identified infectious agents in the analyzed specimens (Fig 4). First, a wide distribution of zoonotic parasites was evidenced in the country. Then, there was an accumulation in the Central Pacific region of specimens with acanthocephaliasis (12 with *Prosthenorchis* spp., one with *Macracanthorhynchus* spp.), and an accumulation of specimens with gastrointestinal nematodes in the great metropolitan area and tourist areas of Guanacaste (six with *Angiostrongylus* spp., one with *Baylisascaris* spp., one with *Ancylostoma* spp.). Additionally, vector-borne diseases occurred exclusively in specimens from coastal regions and altitudes less than 300 meters above sea level (11 with filariae, two with flaviviruses). The CDV in carnivores from various areas of the country did not show a specific distribution pattern (n = 10). The analyzed specimens associated with these infectious agents can be observed in S1 Table.

**Table 5. Number of infectious agents tested and positive in birds according to etiology.**

| Avian taxonomic groups / infectious agent | | Pelecaniformes | Accipitriformes | Anseriformes | Ciconiiformes | Piciformes | Coraciiformes |
|---|---|---|---|---|---|---|---|
| **Viral** | Alphaviruses (n = 3) | 0 | 0 | 0 | 0 | 0 | 0 |
| | Flaviviruses (n = 3) | 2 | 0 | 0 | 0 | 0 | 0 |
| | Influenza virus (n = 9) | 0 | 0 | 0 | 0 | 0 | 0 |
| | Newcastle virus (n = 9) | 0 | 0 | 0 | 0 | 0 | 0 |
| **Bacterial** | *C. perfringens* (n = 1) | 0 | 0 | 0 | 0 | 0 | 0 |
| | *E. coli* (n = 1) | 0 | 0 | 0 | 0 | 0 | 0 |
| | *K. pneumoniae* (n = 1) | 0 | 0 | 0 | 0 | 0 | 0 |
| | *Salmonella* spp. (n = 1) | 0 | 0 | 0 | 0 | 0 | 0 |
| | *S. aureus* (n = 1) | 0 | 0 | 0 | 0 | 0 | 0 |
| **Metazoan parasites [1]** | *Contracaecum* spp. | 2 | 0 | 0 | 0 | 0 | 0 |

n: Number of tested.

[1] only zoonotic metazoan parasites are shown.

## Discussion

The WHMP schemes have proven to be a fundamental tool in monitoring pathogens of zoonotic importance [62–64]. These surveillance systems are even more critical in geographical

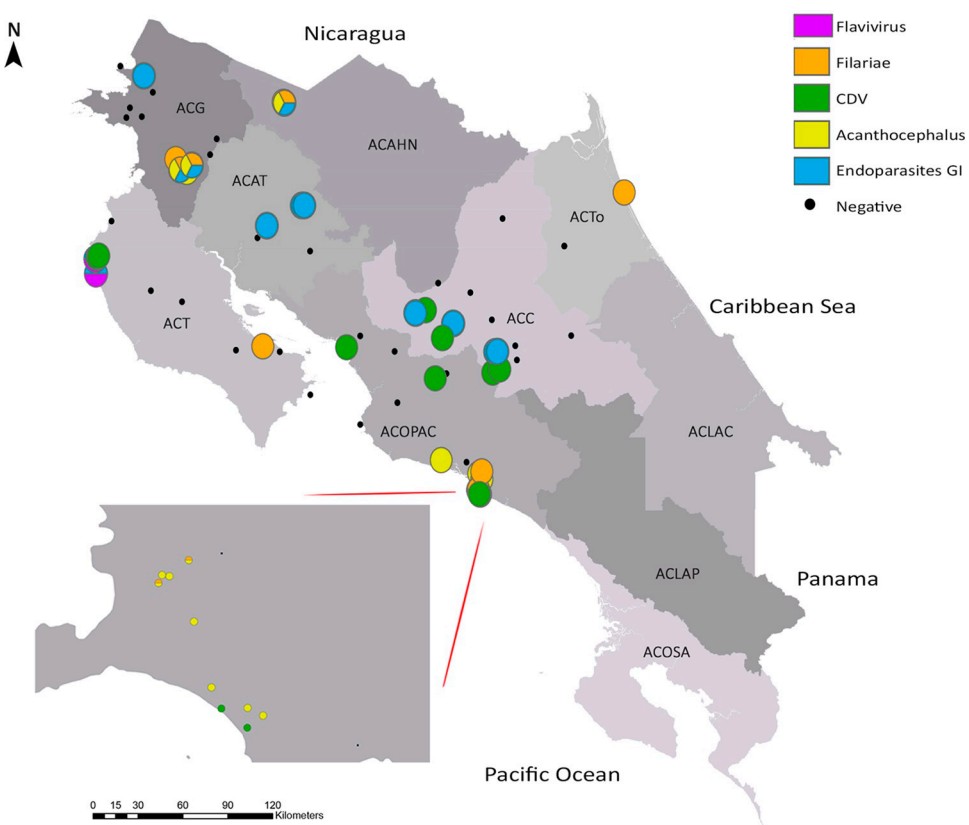

**Fig 4. Geographical distribution of the most frequently identified infectious agents in the referred specimens.** The individuals reported as negative were depicted even though the infectious agent was not detected in the complementary analyzes or no lesions suggestive of the disease were found in the pathological analysis.

areas where high rates of biodiversity are prominent [14]. For example, Costa Rica is economically dependent on its ecotourism services, and its fauna is one of its most important assets [65]. However, no epidemiological surveillance system is currently directed to wildlife to study outbreaks or other health events.

Furthermore, implementing these types of schemes is essential for a country considered a "hotspot" for the appearance or emergence of new infectious agents; however, we encountered some obstacles when performing this study [11,13]. These obstacles are mainly related to lack of legislation for data collection, willingness to cooperate between agencies, financial disincentives and logistical problems for the storage and transport of carcasses, and difficulties similar to those described by some authors [66,67].

The storage capacity and transport logistics directly impact WHMPs. Urban areas with transportation facilities reported and dispatched more carcasses, in contrast to remote or difficult-to-access regions with less participation. These patterns tally with previous reports indicating that notification of wildlife mortality or morbidity generally depends on the initial detection of cases by the general public. Consequently, detected cases are biased towards events in populated or easily accessible areas [17,68,69]. Nevertheless, this shows that the existing logistics in ACC and ACOPAC (urban areas with the highest number of reported cases) can be used to maintain the WHMP at least on a regional level. Likewise, it is necessary to expand the network of laboratories to include other institutions with pathological diagnosis capacity within the WHMP scheme to reduce reliance on the storage and transportation of carcasses. This measure has been shown to improve coverage in distant regions and increase case reporting [19,20,63].

The lack of guidelines and legislation also limited participation and case detection. This means that the system is maintained by the self-interest of officials and interpersonal relationships of people from different institutions. These findings are consistent with previous evaluations of the veterinary services of Costa Rica [26]. To ensure the long-term sustainability of the WHMP, legislation and regulations are necessary to provide financial support and clarify the specific functions of each institution. This could facilitate coordination and cooperation between institutions to notify and transport specimens [19,20,62,63].

The notification and referral of cases relied heavily on the management centers that provide veterinary care to wild animals. Other studies have proposed these institutions as an indispensable tool within the WHMPs due to the large amount of information they can generate for the system [70,71]. The performance of necropsy by the veterinary doctors of management centers would greatly support the efficiency and sustainability of this WHMP, reducing the demand for transportation (only samples would be transported, not complete carcasses), and it would eliminate freezing; facilitating diagnosis. This could encourage the participation of management centers from distant regions without storage capacity, thus increasing coverage. However, the previous regulations, manuals and procedures for post-mortem analysis and sampling procedures are necessary, as occurs with other surveillance schemes, to avoid affecting the diagnosis since pathological analysis is vital in passive surveillance schemes [17,20,21].

Carnivores and primates were the taxa with higher representation. These data can be associated with the fact that they are medium to large-sized animals, more charismatic, and with a more significant contact of these species with human environments, facilitating the recognition of morbidities and mortalities by the population [69]. Therefore, taxa should be prioritized in the WHMPs, since it allows for optimizing the use of resources. Furthermore, in addition to their easy detection, they are taxa in which various pathogenic agents can circulate [5,7].

In contrast, obtaining viable bird carcasses for post-mortem analysis was challenging due to the advanced degree of autolysis, wasting important transport and storage resources, an

obstacle experienced in other studies [72]. Wild birds should be included in surveillance programs for influenza virus and Newcastle disease virus that maintains in the poultry production systems in the country. This would maintain monitoring as is done in other WHMPs without the need for post-mortem analysis, thus avoiding wasted resources [19].

In addition, most cases with a traumatic cause of death presented some pre-existing infectious pathology. Free-living animals are naturally exposed to infectious agents, so it is common to find them in incidental lesions in post-mortem analysis [73–75]. These cases allow the detection of infectious agents in passive surveillance, whether or not they are associated with the cause of death [73,76–78]. Therefore, they must be kept within the cases to be analyzed.

The diagnostic capability allowed the WHMP to detect infectious agents that could affect the health of domestic animals, public health, and the conservation of wild species. For example, our study shows the presence of potentially zoonotic bacterial infectious agents classified as emerging diseases in some regions [79–81]. The most relevant are *Klebsiella pneumoniae*, *Escherichia coli*, and *Staphylococcus aureus*, which were associated with primary disease in some of the analyzed specimens. In addition, these bacteria currently top the list of infectious agents with antibiotic resistance genes, thus showing the importance of monitoring these agents in WHMP schemes [28,82–84].

We also detect vector-borne diseases, which are recognized as agents with epidemic potential in Latin America due to tropical regions' climatic, health, and socioeconomic conditions that favor their spread [85–87]. We identified primates, carnivores, and birds with infectious agents of vector transmission, for example, *Dirofilaria* spp., *Dipetalonema* spp., and flavivirus mainly present by the coast. Most of these cases come from regions already defined as endemic areas for these infectious agents in domestic animals, which reveals a possible transmission by this route and a potential risk for the conservation of the species [88–91].

Detection of these vector-borne pathogens also reveals a potential risk to public health in places with a high rate of tourists visiting Costa Rica. This risk is reinforced by health system reports showing at least three disease cases in humans associated with *Dirofilaria immitis* and isolated cases of subcutaneous filariasis [92–94]. Furthermore, detecting virus-related mortalities such as West Nile in wild birds (as was possibly our case) allows early alerts. It has been shown that there is a higher risk of exposure for human populations close to the regions where mortalities of wild birds occur [95,96].

The CDV was frequently detected in our study, reflecting the relevance of this virus in the role of spillover towards carnivore species and possibly the implications of a spillback towards susceptible or non-vaccinated domestic canines [97–99]. Endemic CDV outbreaks have been reported anecdotally throughout Costa Rica and America in dog populations. More recently, sporadic outbreaks in wild carnivores of urban and suburban areas have been recorded [97,100]. Unfortunately, Costa Rica does not have official data on the domestic dog population. Therefore, herd immunity data in this population is uncertain, especially for dogs without an owner or in non-urban areas. This poses a risk to wild carnivores, especially in urban areas with susceptible canine populations. Furthermore, the possibility of transmission of this virus to other species beyond carnivores is a hypothesis that has been investigated [101]. Given the high diversity of vertebrates in Costa Rica and the high circulation of CDV detected, this virus should be considered within epidemiological surveillance programs.

Also, this study's gastrointestinal and pulmonary metazoan parasites are relevant for public health and wildlife conservation programs. For instance, we detect the nematodes *Angiostrongylus* spp., *Baylisascaris* spp., *Ancylostoma* spp., and *Cylicospirura* spp. in mammalian species located in densely populated areas. In addition, we detected cases with acanthocephalans (*Prosthenorchis* spp., *Macracanthorhynchus* spp.) concentrated in the Central Pacific region and parasites transmitted by water or aquatic food such as the cestode *Spirometra* spp. in the

country's northern region. This result proves a cost-effective tool for the WHMP, which does not require financial resources beyond qualified personnel for the morphological identification of worms and allows the detection of pathogenic agents that primarily impact children [102–104].

This study did not detect rabies virus infections. These findings are supported by previous studies on wild animals in Costa Rica [29]. However, this passive surveillance program allowed expanding coverage in the number of species and geographic regions through constant monitoring of wild species. Human and livestock fatalities have been reported associated with rabies infections, which stresses the relevance of its continuous monitoring of species that can act as reservoirs [35,105]. A similar situation applies to Newcastle and Influenza virus. In our samples, none of the birds showed evidence of disease or associated clinical signs; however, due to the great relevance of these diseases to the country and the risk for national production, it is advisable to establish routine monitoring by the animal health agency under the WHMP scheme [35,106]. Including serological monitoring of tuberculosis and brucellosis of wild species in the scheme would even be advisable. These national epidemiological surveillance programs already include some wild species, and coverage could be expanded [107].

Finally, we could not identify the cause of death in some of the samples analyzed. Although we tested for the main circulating infectious agents in Costa Rica, no conclusive data was obtained. Ranges of 17–22% have been reported in pathological studies in wild species, where the causative agent of the disease cannot be determined, mainly associated with the degree of autolysis and the diagnostic complexity [68,74,76]. These results are consistent with the percentages of an absence of identification of the etiological agent in our samples. Although proving that diagnostic capacity is acceptable, further work is necessary to develop robust diagnostic techniques for wild animal testing. Further efforts and incentives, financed by government authorities, are required for pathogen surveillance in wildlife through the consistent implementation of tools such as new generation metagenomics [108–111].

Although the proposed program is limited to the country's resources and infrastructure, and it is clear that it is not generally applicable, it is important to start evaluating the implementation of these programs in regions where disease surveillance in wildlife is minimal. For example, this study shows that this passive surveillance scheme is cost-effective and feasible to establish in countries with limited resources. Furthermore, this scheme was possible since we could adapt the infrastructure dedicated to monitoring diseases in productive animals according to the scope and objectives of monitoring wildlife specific to each region. We also showed sufficient diagnostic capacity in the country for detecting infectious agents of zoonotic and conservation importance in wild animals. If this scheme is maintained over time, it will generate data to allow the decision-making to promote the conservation of species, animal health, and public health by knowing the circulation and behavior of these pathogens [68].

This study highlights the need for an inter-institutional and trans-institutional commitment to the sustainability over time of this surveillance scheme. Participant institutions must remain motivated and focused on the benefits beyond the economic part. The feedback to field staff and the frequent reports of the importance of detected pathogens are crucial to maintaining motivation and detection network, as in our case. In addition, the information generated from the experience of the initial establishment of a WHMP is critical to meeting the challenges involved in developing this type of scheme in regions with limited resources and established as hotspots for emerging infectious diseases [13,112]. Although it is necessary to standardize methods and techniques for monitoring pathogens in wildlife, the development of pilot schemes allows sharing experiences with programs already installed and leads to subsequent optimization and standardization studies that will facilitate the exchange of information and expand coverage [112].

## Supporting information

**S1 Table. Biological data and more representative pathological findings of the analyzed specimens.**
(DOCX)

## Acknowledgments

The authors express deep appreciation to the National Animal Health Service (SENASA) and National Wildlife Service (SINAC) for their support and cooperation in this study and the wildlife management centers for the notification of specimens, especially DVM. Isabel Hagnauer Barrantes from Zooave rescue center, DVM. Martha Cordero Salas from Las Pumas rescue center and DVM. For their particular interest and participation, Carmen Soto Valverde from the Kids Saving the Rainforest rescue center. Also, the collaboration provided by the technicians of the laboratories of the Escuela de Medicina Veterinaria of the Universidad Nacional is appreciated.

## Author Contributions

**Conceptualization:** Alejandro Alfaro-Alarcón.

**Data curation:** Fernando Aguilar-Vargas, Tamara Solorzano-Scott, Mario Baldi, Alejandro Alfaro-Alarcón.

**Formal analysis:** Fernando Aguilar-Vargas, Tamara Solorzano-Scott, Mario Baldi, Alejandro Alfaro-Alarcón.

**Funding acquisition:** Alejandro Alfaro-Alarcón.

**Investigation:** Fernando Aguilar-Vargas, Tamara Solorzano-Scott, Alejandro Alfaro-Alarcón.

**Project administration:** Alejandro Alfaro-Alarcón.

**Resources:** Fernando Aguilar-Vargas, Tamara Solorzano-Scott, Mario Baldi, Elías Barquero-Calvo, Ana Jiménez-Rocha, Carlos Jiménez, Marta Piche-Ovares, Gaby Dolz, Bernal León, Alejandro Alfaro-Alarcón.

**Software:** Fernando Aguilar-Vargas, Tamara Solorzano-Scott, Mario Baldi.

**Supervision:** Alejandro Alfaro-Alarcón.

**Visualization:** Fernando Aguilar-Vargas, Tamara Solorzano-Scott, Alejandro Alfaro-Alarcón.

**Writing – original draft:** Fernando Aguilar-Vargas, Tamara Solorzano-Scott, Mario Baldi, Elías Barquero-Calvo, Alejandro Alfaro-Alarcón.

**Writing – review & editing:** Fernando Aguilar-Vargas, Tamara Solorzano-Scott, Mario Baldi, Elías Barquero-Calvo, Ana Jiménez-Rocha, Carlos Jiménez, Marta Piche-Ovares, Gaby Dolz, Bernal León, Eugenia Corrales-Aguilar, Mario Santoro, Alejandro Alfaro-Alarcón.

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
