## [Decision Letter · Decision Letter 0]

30 Mar 2022

PONE-D-21-39637Passive epidemiological surveillance in wildlife in Costa Rica identifies pathogens of zoonotic and conservation importancePLOS ONE

Dear Dr. Alfaro-Alarcon,

Thank you for submitting your manuscript to PLOS ONE. After careful consideration, we feel that it has merit but does not fully meet PLOS ONE’s publication criteria as it currently stands. Therefore, we invite you to submit a revised version of the manuscript that addresses the points raised during the review process.

Both reviewers note a disconnect between the stated aims of the paper (to evaluate the feasibility of implementing a passive surveillance system) and the actual content of the paper (results of pathological findings and their relevance for conservation/public health). Both reviewers in turn suggest the authors provide substantially more attention to a forward-looking component of their manuscript and propose concrete recommendations to implement such a program and how to make it sustainable/reproducible (one reviewer also suggests that the manuscript would greatly benefit from a flowchart of methodologies). Both reviewers also provide a number of more minor suggestions to improve clarity and precision of the manuscript. Some important clarifying points are also noted about inclusion/exclusion criteria for animals from rescue centers, given the possibility for pathogen exposure within the centers themselves (rather than from the forests).==============================

We look forward to receiving your revised manuscript.

Kind regards,

Daniel Becker

Academic Editor

PLOS ONE

Journal Requirements:

2. We note that Figure 3 in your submission contain map images which may be copyrighted. All PLOS content is published under the Creative Commons Attribution License (CC BY 4.0), which means that the manuscript, images, and Supporting Information files will be freely available online, and any third party is permitted to access, download, copy, distribute, and use these materials in any way, even commercially, with proper attribution. For these reasons, we cannot publish previously copyrighted maps or satellite images created using proprietary data, such as Google software (Google Maps, Street View, and Earth). For more information, see our copyright guidelines: http://journals.plos.org/plosone/s/licenses-and-copyright.

1. You may seek permission from the original copyright holder of Figure 3 to publish the content specifically under the CC BY 4.0 license.  

Reviewers' comments:

Reviewer's Responses to Questions

**Comments to the Author**

1. Is the manuscript technically sound, and do the data support the conclusions?

Reviewer #1: Partly

Reviewer #2: Yes

2. Has the statistical analysis been performed appropriately and rigorously? 

Reviewer #1: N/A

Reviewer #2: N/A

3. Have the authors made all data underlying the findings in their manuscript fully available?

Reviewer #1: Yes

Reviewer #2: Yes

4. Is the manuscript presented in an intelligible fashion and written in standard English?

Reviewer #1: Yes

Reviewer #2: Yes

5. Review Comments to the Author

Reviewer #1: The study addresses a critical gap in wildlife disease surveillance by implementing a useful and needed tool to survey wild animals passively. They highlight the urgency of this epidemiological surveillance system, with an extensive pathological work paired with a broad literature review on the topic. The authors show the relevance of such approach with significant examples such as the circulation of Canine Distemper Virus, which could potentially be a treat for wildlife and domestic dogs. However, although the authors state that they aim “to evaluate the technical and infrastructural feasibility to establish” a passive surveillance system, they are only demonstrating the feasibility of this approach by describing in detail they pathological findings and discussing the relevance of their results for public health and conservation, without any further analysis or clear evaluation of viability of their program.

The information their research provided is however, of great value, and the authors make a strong case of the need of such passive surveillance in a low-resource country like Costa Rica. They showed they have the infrastructural and expertise to carry this surveillance program, but probably lack the financial and logistical support to make this program permanent. However, they don’t seem to propose a protocol or guidelines on how to make this program available and reproducible in the country (including the sites they identify as under-studied) and other regions. For instance, they could address aspects such as that most of their samples came from wildlife rescue center as possible ways of improving and sustaining their surveillance program. In line 278-280 they even mentioned that they encounter obstacles, but do not mention how those obstacles were overcome. I suggest they proposed clearer steps moving forward, if they truly want to evaluate the implementation of this valuable wildlife surveillance program.

The text has some grammatical errors and redundant sentences, that makes the reading hard to follow. I mention some of these issues below, in addition to other observations.

Lines 50-53. Covid-19 would not make a good example. Although it may have a zoonotic origin, it is considered a human-to-human disease. May be consider another example.

Lines 54-56. These two sentences seem redundant. Consider rephrasing.

Line 59. Consider changing “countries” for “regions”, and eliminating the “(including Costa Rica)”

Line 61. Consider changing “geographic regions” for “countries”

Line 64. Do you mean increased land use change?

Lines 76-77. In that case, since they are post-mortem examinations, it would not be morbidity, right?

Line 77-79. I don’t follow this sentence. What do you mean by “obtaining a sustainable tool that allows understanding the emerging potential of different pathogens.”? Also, consider changing “profitability” to “cost-effective” because the latter has a connotation of lucrative.

Line 88. A disease is not a synonym for pathogen, please change “vector-borne diseases” accordingly.

Lines 89-93 and 96-98. Please rephrase these sentences, as they are unclear and hard to follow.

Lines 99-101. Avoid giving specific results here (e.g., 85 carcasses), just general and main findings.

Line 108. Isn’t the permit R-SINAC-PNI-ACLAC-039 for the Area de Conservacion La Amistad Caribe only?

Table 1. superscript b. Cloacal swab is not a tissue, please change accordingly.

Lines 167-168. Please change “specimens of the birds and 76 (89.5%) of the mammals” to “bird and 76 (89.5%) mammal specimens”

Lines 170-173. This information doesn’t seem to belong in this subsection but rather in either of the following ones.

Lines 178-181. 24 + 3 + 1 + 4 = 32, not 31 as stated below. Please check this. Also, please be specific on what are you referring to with “parasites”.

Lines 184-193. I am confused with this whole paragraph and the data presented below (including in Table 2). I imagine that you have several tissue samples from the same carcass, because you had 85 individuals total, but are now presenting 266 lesions total (199 + 67). Please clarify.

Lines 195-196. 11 + 5 + 4 = 20, it doesn’t add to 21. Please check this.

Lines 199-200. Do you mean that the rest of the tissues with infection lesions were associated with a pathogen? And what is the complementary analysis?

Table 2. I suggest you change “Bird” for “Aves”, to keep the correct taxa name for all the groups sampled.

Lines 241-242. Do you mean in here that all the birds, total of nine, were screened for viruses? From which, only the two that were involved in a mass mortality event were positive for flaviviruses? Please rephrase accordingly.

Line 300. Rephrase to: Regarding mammals, carnivores and primates were the taxa with higher representation

Lines 302-304. It is unclear what do you mean with this sentence. Are you suggesting that human proximity makes transmission of this infectious agents more common, or that the sole proximity facilitates pathogen detection? Also, it could be possible that cases of more charismatic animals are over-represented.

Lines 305-306. Anthropogenic effect it’s a broad term. Can you rephrase and be more specific?

Lines 313-315. What do you mean by human conditions? Also, I am not sure if you are using the term “evolve” correctly here. Do you mean pathogen evolution or how the disease develops?

Lines 331-332. This sentence is very vague. Please rephrase and be more specific.

Line 335. Please change “(CoastLine)” to “by the coast”.

Lines 336-340. What about conservation implications? What if this pathogen detection reveals the transmission of pathogens from domestic animals to wildlife?

Line 349 (and others) Please be consistent with the naming. If you used “CDV” at the beginning after explaining what it stands for, you should keep using the same name. Also, include the genus information at the beginning, where you mentioned the pathogens you are going to be screening for.

Line 353. Please change “studies” to “study”.

Lines 355-356. Can you provide a reference for this estimation?

Line 363. With “productive animal” do you mean livestock?

Line 379. Replace “the Nasua narica species” to “Nasua narica”

Figure 3. Giving that you mentioned that samples are biased toward the wildlife management centers, it would be informative to show in the map the location of the centers that sent the carcasses. If there is no clear purpose for the color coding on the map (I imagine indicating elevation), I suggest removing it, because it makes the figure harder to interpret. Finally, consider using other type of icons to show the pathogens detected that are not 3D.

Figure S1. This map was very noisy and hard to interpret. Consider simplifying as much as possible. May be to order, instead of genus, and removing the elevation color.

Please check your references. Scientific names should be in italics and there are a few mistakes, such as in Line 534, where Panama and Costa Rica are not capitalized.

Reviewer #2: Initially I would like to congratulate Dr. Alfaro-Alarcon for this manuscript, we know that research efforts to develop studies like the one presented here require a network of motivated institutions and people, this can be evidenced in this manuscript. So please feel proud of this manuscript fruit of his continuous work for almost three years. Congratulations.

6. PLOS authors have the option to publish the peer review history of their article (what does this mean?). If published, this will include your full peer review and any attached files.

Reviewer #1: No

Reviewer #2: **Yes: **PEDRO ENRIQUE NAVAS SUÁREZ

---

## [Author Response · Author response to Decision Letter 0]

1 Jun 2022

Rebuttal Letter

PONE-D-21-39637

Passive epidemiological surveillance in wildlife in Costa Rica identifies pathogens of zoonotic and conservation importance by Alfaro-Alarcón et al.

PLOS ONE

Dear reviewers:

General comments: 

Initially, we would like to thank you for the positive and accurate comments you have made. Undoubtedly, these changes substantially improve the information and scientific relevance of this manuscript. Previously, we believed that we had addressed the aspects associated with the feasibility of implementing a passive epidemiological surveillance system for diseases in wild animals in Costa Rica. However, with these changes, the arguments are more direct with each point addressed and broader in the improvements needed for this system to be sustainable over time and reproducible in other regions, obtaining the important benefits generated by monitoring the health of wild animals, as exemplified in this research.

Academic Editor

Both reviewers note a disconnect between the stated aims of the paper (to evaluate the feasibility of implementing a passive surveillance system) and the actual content of the paper (results of pathological findings and their relevance for conservation/public health). Both reviewers in turn suggest the authors provide substantially more attention to a forward-looking component of their manuscript and propose concrete recommendations to implement such a program and how to make it sustainable/reproducible (one reviewer also suggests that the manuscript would greatly benefit from a flowchart of methodologies).

Response: These changes were accepted. The presentation of the results and the discussion of the arguments were reformulated, with the purpose to be much more prospective and linked to the objective of this study. A flowchart was added on the proposed surveillance system to make this reproducible.

Both reviewers also provide a number of more minor suggestions to improve clarity and precision of the manuscript. Some important clarifying points are also noted about inclusion/exclusion criteria for animals from rescue centers, given the possibility for pathogen exposure within the centers themselves (rather than from the forests).

Response: These changes were accepted. The answer is given at every minor suggestion.

Response: The manuscript was adjusted to the style requirements established by PlosONE according to the indicated templates.

2. We note that Figure 3 in your submission contain map images which may be copyrighted. All PLOS content is published under the Creative Commons Attribution License (CC BY 4.0), which means that the manuscript, images, and Supporting Information files will be freely available online, and any third party is permitted to access, download, copy, distribute, and use these materials in any way, even commercially, with proper attribution. For these reasons, we cannot publish previously copyrighted maps or satellite images created using proprietary data, such as Google software (Google Maps, Street View, and Earth). For more information, see our copyright guidelines: http://journals.plos.org/plosone/s/licenses-and-copyright.

Response: The maps submitted as figures in this manuscript are not copyrighted. The maps were created by the researchers for this study, using ArcGIS Desktop 10.8.1 software, license type: advanced, paid for by the National University of Costa Rica.

Response: The correct information has been sent.

Reviewer #1: 

The study addresses a critical gap in wildlife disease surveillance by implementing a useful and needed tool to survey wild animals passively. They highlight the urgency of this epidemiological surveillance system, with an extensive pathological work paired with a broad literature review on the topic. The authors show the relevance of such approach with significant examples such as the circulation of Canine Distemper Virus, which could potentially be a treat for wildlife and domestic dogs. However, although the authors state that they aim “to evaluate the technical and infrastructural feasibility to establish” a passive surveillance system, they are only demonstrating the feasibility of this approach by describing in detail they pathological findings and discussing the relevance of their results for public health and conservation, without any further analysis or clear evaluation of viability of their program.

The information their research provided is however, of great value, and the authors make a strong case of the need of such passive surveillance in a low-resource country like Costa Rica. They showed they have the infrastructural and expertise to carry this surveillance program, but probably lack the financial and logistical support to make this program permanent. However, they don’t seem to propose a protocol or guidelines on how to make this program available and reproducible in the country (including the sites they identify as under-studied) and other regions. For instance, they could address aspects such as that most of their samples came from wildlife rescue center as possible ways of improving and sustaining their surveillance program. In line 278-280 they even mentioned that they encounter obstacles, but do not mention how those obstacles were overcome. I suggest they proposed clearer steps moving forward, if they truly want to evaluate the implementation of this valuable wildlife surveillance program.

Response: The presentation about the results and discussion of the arguments was reformulated. In the current manuscript, the obstacles that the country faces for the implementation of a vigilance system in wild animals are developed. At the same time, we included recommendations that could be useful to overcome these difficulties and facilitate permanent implementation.

The text has some grammatical errors and redundant sentences, that makes the reading hard to follow. I mention some of these issues below, in addition to other observations.

Lines 50-53. Covid-19 would not make a good example. Although it may have a zoonotic origin, it is considered a human-to-human disease. May be consider another example.

Response: It is exemplified by covid-19 as a pathogen with possible origin in wild animals that can generate outbreaks. But this change was accepted. The change appears on lines 50-53: 

“Examples of how these diseases can impact public health, animal health and wildlife have been the recent outbreaks of yellow fever and West Nile virus, which show the need to have infrastructure and diagnostic capacity to ensure a constant surveillance of potentially zoonotic agents.”

Lines 54-56. These two sentences seem redundant. Consider rephrasing.

Response: This change was accepted. The change appears on line 54: “Wildlife populations act as reservoirs and can play various roles in the epidemiology of numerous pathogens.”

Line 59. Consider changing “countries” for “regions”, and eliminating the “(including Costa Rica)”

Response: This change was accepted. The change appears on line 58.

Line 61. Consider changing “geographic regions” for “countries”

Response: This change was accepted. The change appears on line 60.

Line 64. Do you mean increased land use change?

Response: Yes, the change appears on line 63.

Lines 76-77. In that case, since they are post-mortem examinations, it would not be morbidity, right?

Response: Yes, morbidity was removed. The change appears on line 74. 

Line 77-79. I don’t follow this sentence. What do you mean by “obtaining a sustainable tool that allows understanding the emerging potential of different pathogens.”? Also, consider changing “profitability” to “cost-effective” because the latter has a connotation of lucrative.

Response: This sentence was removed. 

Line 88. A disease is not a synonym for pathogen, please change “vector-borne diseases” accordingly.

Response: This change was accepted. The change appears on line 87.

Lines 89-93 and 96-98. Please rephrase these sentences, as they are unclear and hard to follow.

Response: These sentences were rephrased. The change appears on lines 88-90: 

“This reflects the urgency of establishing a permanent WHMP, where aspects such as general health status and monitoring of zoonotic pathogens in wildlife are considered, facilitating knowledge of the ecoepidemiology of these agents at the local level.”

Lines 99-101. Avoid giving specific results here (e.g., 85 carcasses), just general and main findings.

Response: This change was accepted. The change appears on lines 95-99. 

Line 108. Isn’t the permit R-SINAC-PNI-ACLAC-039 for the Area de Conservacion La Amistad Caribe only?

Response: Yes, by mistake we only included one. The change appears on lines 102-106: The study was approved by the Ministry of Environment and Energy (MINAE) (wildlife authority) through permits (R-SINAC-PNI: -ACAT-040, ACAHN-18, ACTo-022, ACT-OR-DR-43, ACG-026, ACLAC-039, ACLAP-023, ACOPAC-005, ACC-037), and with the support of the animal health authority, the National Animal Health Service through the office (SENASA-DG-0277-18).

Table 1. superscript b. Cloacal swab is not a tissue, please change accordingly.

Response: The change appears on the Table 1. “Lung and Trachea tissue and cloacal swab.”

Lines 167-168. Please change “specimens of the birds and 76 (89.5%) of the mammals” to “bird and 76 (89.5%) mammal specimens”.

Response: These sentences were rephrased. The change appears on lines 179-184: 

“According to the taxonomic order, we received: 29.4% (25/85) Carnivora, 29.4% (25/85) Primate, 12.9% (11/85) Pilosa, 5.9% (5/85) Didelphimorphia, 4.7% (4/85) Rodentia, 4.7% (4/85) Artiodactyla, 2.3% (2/85) Cingulate, 2.3% (2/85) Pelecaniformes, 2.3% (2/85) Accipitriformes, 2.3% (2/85) Anseriformes, 1.2% (1/85) Ciconiiformes, 1.2% (1/85) Piciformes and 1.2% (1/85) Coraciiformes.”

Lines 170-173. This information doesn’t seem to belong in this subsection but rather in either of the following ones.

Response: This sentence was removed.

Lines 178-181. 24 + 3 + 1 + 4 = 32, not 31 as stated below. Please check this. Also, please be specific on what are you referring to with “parasites”.

Response: These data were reviewed, certainly there are 32. The change appears on lines 196-198: “Of the individuals with a cause of traumatic death, 67.4% (32/85) concomitantly presented some infectious agent with or without associated disease (24 with gastrointestinal and pulmonary parasitic worms, three with bacteria, one with protozoa and four with multiple microorganisms).”

Lines 184-193. I am confused with this whole paragraph and the data presented below (including in Table 2). I imagine that you have several tissue samples from the same carcass, because you had 85 individuals total, but are now presenting 266 lesions total (199 + 67). Please clarify.

Response: Yes, lesions of different tissues were identified in each carcass. However, this paragraph and Table 2 were removed.

Lines 195-196. 11 + 5 + 4 = 20, it doesn’t add to 21. Please check this.

Response: This paragraph was removed.

Lines 199-200. Do you mean that the rest of the tissues with infection lesions were associated with a pathogen? And what is the complementary analysis?

Response: This paragraph was removed.

Table 2. I suggest you change “Bird” for “Aves”, to keep the correct taxa name for all the groups sampled.

Response: Table 2 includes the correct taxonomic name at the order level of the different groups sampled.

Lines 241-242. Do you mean in here that all the birds, total of nine, were screened for viruses? From which, only the two that were involved in a mass mortality event were positive for flaviviruses? Please rephrase accordingly.

Response: This sentence was rephrased. The change appears on line 230: 

“All birds submitted were evaluated for virus presence (n=9), two of these were positive for flaviviruses.”

Line 300. Rephrase to: Regarding mammals, carnivores and primates were the taxa with higher representation

Response: The change appears on line 291.

Lines 302-304. It is unclear what do you mean with this sentence. Are you suggesting that human proximity makes transmission of this infectious agents more common, or that the sole proximity facilitates pathogen detection? Also, it could be possible that cases of more charismatic animals are over-represented.

Response: This paragraph was rephrased. The change appears on lines 291-295: 

“Carnivores and primates were the taxa with higher representation. These data can be associated with the fact that they are medium to large-sized animals, more charismatic and with a more significant contact of these species with human environments, facilitating the recognition of morbidities and mortalities by the population. These taxa should be prioritized in the WHMPs, since it allows to optimize the use of resources because, in addition to their easy detection, they are taxa in which various pathogenic agents can circulate.”

Lines 305-306. Anthropogenic effect it’s a broad term. Can you rephrase and be more specific?

Response: This paragraph was removed.

Lines 313-315. What do you mean by human conditions? Also, I am not sure if you are using the term “evolve” correctly here. Do you mean pathogen evolution or how the disease develops?

Response: Yes, reference was made to how the disease develops. However, this paragraph was removed.

Lines 331-332. This sentence is very vague. Please rephrase and be more specific.

Response: These sentences were rephrased. The change appears on lines 315-316: 

“We also detect vector-borne diseases, which are recognized as agents with epidemic potential in Latin America due to the climatic, health and socioeconomic conditions of tropical regions that favor their spread.”

Line 335. Please change “(CoastLine)” to “by the coast”.

Response: The change appears on line 318. 

Lines 336-340. What about conservation implications? What if this pathogen detection reveals the transmission of pathogens from domestic animals to wildlife?

Response: It is a risk to public health and to the conservation of species. The cases were in endemic regions with high transmission of these pathogens in domestic animals. This is mentioned on line 318-320.

Line 349 (and others) Please be consistent with the naming. If you used “CDV” at the beginning after explaining what it stands for, you should keep using the same name. Also, include the genus information at the beginning, where you mentioned the pathogens you are going to be screening for.

Response: The change was accepted. This was corrected throughout the manuscript.

Line 353. Please change “studies” to “study”.

Response: This change was accepted. The change appears on line 327.

Lines 355-356. Can you provide a reference for this estimation?

Response: No, it was corrected on lines 331-332: 

“Costa Rica does not have official data on the domestic dog population”.

Line 363. With “productive animal” do you mean livestock?

Response: This change was accepted. The change appears on line 348

Line 379. Replace “the Nasua narica species” to “Nasua narica”

Response: This sentence was removed.

Figure 3. Giving that you mentioned that samples are biased toward the wildlife management centers, it would be informative to show in the map the location of the centers that sent the carcasses. If there is no clear purpose for the color coding on the map (I imagine indicating elevation), I suggest removing it, because it makes the figure harder to interpret. Finally, consider using other type of icons to show the pathogens detected that are not 3D.

Figure S1. This map was very noisy and hard to interpret. Consider simplifying as much as possible. May be to order, instead of genus, and removing the elevation color.

Response: The figures corresponding to the maps were edited. The Color coding shows the different conservation areas; however, this was previously not specified. The location of the participating rescue centers was included.

Please check your references. Scientific names should be in italics and there are a few mistakes, such as in Line 534, where Panama and Costa Rica are not capitalized.

Response: References were checked and corrected.

Reviewer #2: 

General comments: Initially I would like to congratulate Dr. Alfaro-Alarcon for this manuscript, we know that research efforts to develop studies like the one presented here require a network of motivated institutions and people, this can be evidenced in this manuscript. So please feel proud of this manuscript fruit of his continuous work for almost three years. Congratulations.

This MS is a bit difficult to evaluate, since it could have two different perspectives:

1. Formulation and implementation of the pilot of a Wildlife Health Monitoring Program (WHMP).

2. A study of causes of death and main pathological processes in various species of vertebrates.

My review is in order to analyze this MS with the postulating of WHMP. I strongly believe that this is a manuscript that with some revisions can be used as an example for the installation of a wildlife health surveillance program. Below I respectfully highlight some points that should be considered by you. 

I initially consider that some infectious agents should be considered as "primary" for monitoring. I believe that the rabies virus and the flaviviruses would be good pathogens in mammals and influenza and the New Castle virus in birds. These diseases must be carefully selected since they will be the ones that will generate the demand for a national program by the responsible agency. For the other pathogens, I consider appropriate the use of complementary techniques based on histopathological analysis (CDV, bacterial).

In order to replicate this program, I felt a huge lack of a flowchart in the methodology describing each of the institutions with their respective activities. 

Response: “Primary” agents were established for monitoring, but it was not indicated. The flowchart specifies the surveillance scheme for the different infectious agents (microbiological diagnosis based on pathological findings or routine diagnostics), and the role of each participating institution.

I understand that all the necropsies were carried out by the veterinary pathologists of the university involved in this study, but in order to improve the efficiency and effectiveness of this program, they do not believe that it would be better if the necropsies and sample collection were carried out directly in the institution (in this way the freezing of corpses would be avoided, which improves the histopathological evaluation).

Response: This is one of the measures proposed to improve the sustainability of the WHMP. This change appears on lines 281-290.

The carcasses that were found in the "forest" can really provide insight into the pathogens that are circulating in the ecosystem. However, animals that have entered a rescue center may be exposed to pathogens that are stabilized on the installations. Considering this, was there any inclusion criteria for animals from rescue centers, for example, animals that arrived dead or that lasted a maximum of 24 hours in the institution?

Response: Exclusion criteria were not mentioned above. Lines 112-114 included the previously established exclusion criteria: 

“Carcasses of animals that remained more than 48 hours in the management centers before death were excluded also those that received medication, and those carcasses that were frozen for more than a week when it was necessary to store them.”

I think that seeking to create a diagnostic algorithm within the wildlife health program that you propose, perhaps a classification of causes of death like this could be better:

• Death associated with an infectious agent.

• Death not associated with an infectious agent, with a pre-existing infectious disease.

• Death not associated with an infectious agent, with a molecular diagnosis of an infectious pathogen.

• Death not associated with an infectious agent.

• Undetermined death.

With these categories you could see which species may be sensitive (which would be ideal for passive surveillance by acting as sentinels) and which are resistant (which would be ideal for serological inquiries) to the selected pathogens. I believe that in order to establish a wildlife health surveillance program, particularly in countries with limited resources, it is important to prioritize efforts, to those pathogens of greatest importance to human health (based on epidemiological data from the Health Agency), as well as for biodiversity conservation (based on wildlife mortality data). 

Response: Thanks for the observation. This algorithm was used. This information appears in Table 3.

I confess that when presenting absolute and proportional results I find it interesting to use the following form: X% (n/N). It is not something that they should replicate but if you find it interesting it would be great.

Response: This change was accepted. The data is displayed in this format.

Introduction: Despite being a somewhat extensive introduction (for me), I consider that addressing the points that will be dealt with in the manuscript, I make some specific comments in the minor revision section.

MM: If you accept my comments please describe the WHMP better, I have no objection to the proposed diagnostic methodologies. I am sure that you are quite judicious about it.

Response: To better describe the WHMP, a flowchart has been included detailing the role of each participating institution and the communication of the results obtained.

Results and discussion: As things change in the methodology you will see that the results can be written in another way, I will make two suggestions for tables, and please when you are writing the results do not forget that the focus of the manuscript is not pathology, it is the implementation of the WHMP. Please, be a little more summarized in the pathological aspects (S2 table is enough) and increase the information about the program. I make this same comment for discussion. Remember that discussing whether or not the non-infectious cause of death is frequent, for the purpose of this manuscript, may be secondary. But discussing the limitations, benefits, diagnostic techniques, costs, can be much more interesting.

Tables: 

If the authors consider my suggestions, Table 1 should include the absolute and relative values of the causes of death for each of the taxonomic groups, that is:

Cause of Death / Taxon DAIA DNAIA-PD DNAIA-ADT DNAIA UD

Mammals % (n/N) % (n/N) % (n/N) % (n/N) % (n/N)

Carnivores % (n/N) % (n/N) % (n/N) % (n/N) % (n/N)

Birds % (n/N) % (n/N) % (n/N) % (n/N) % (n/N)

Pelecaniformes % (n/N) % (n/N) % (n/N) % (n/N) % (n/N)

DAIA= Death associated with an infectious agent; DNAIA-PD= Death not associated with an infectious agent, with a pre-existing infectious disease; DNAIA-ADT= Death not associated with an infectious agent, with an additional diagnostic tool; DNAIA= Death not associated with an infectious agent; UD= Undetermined death.

Response: These changes were accepted, and due to the new way of presenting the results, there is a new Table 2 with new information. The pathological aspects were considerably reduced, the table 2 and the paragraphs that indicated the results by lesions were removed. The same was done with the discussion. This was a great observation to improve the quality of the manuscript with respect to WHMP. 

Table 1 include the pathogens diagnosed with the respective techniques. 

Table 3 was included for additional information about the program. 

Tables 4 and 5 were already in the manuscript. Agents detected by birds and mammals were divided because it is impossible to include them all in a single table.

S1. Table: please add the scientific name of all specimens.

Response: This change appears in table S1.

Figures: 

Please homogenize the formatting of the figures, if you are going to place a scale bar, place them in all of them, otherwise, remove them in all of them. In the case of metazoan parasites, describe if they were identified morphologically and/or molecularly, otherwise if they were by tissue characteristics, mention which guide was used (eg, Gardiner or Chitwood), and describe the order and/or the superfamily, and later if they want they can mention the potential genus (explaining that this genus is postulated by previous accounts in the species or close taxonomic group). I confess that I feel a lack of macroscopic photos. but they are not necessary

Response: This change was accepted, all scale bars were eliminated. Also, after the histopathological description of each microphotograph, the diagnostic technique used is indicated.

Figure 1.

A. To exemplify the cases of CDV in none of the animals were inclusion corpuscles observed? If these were observed, my suggestion is to place a photo of one of them. The caption provided describes the image well.

Response: The purpose of this figure was to show inflammatory-type lesions. However, with the reduction of the pathological component and with the reformulation of the results. This figure was removed.

B. The inflammatory component is not fully observed in the microphotograph, I confess that it was a bit difficult for me to show the granulomatous component (If it is possible to take another microphotograph where it is more evident, it is highly recommended), otherwise, I recommend modifying the legend of this Photo.

Response: With the reduction of the pathological component and with the reformulation of the results. This figure was removed.

C. In this microphotography it is not possible to show the Trypanozoma amastigotes, the myocardial inflammatory process is well described in the legend, my suggestion is to place a microphotograph at a higher magnification (eg 400x or 1000x) only of the amastigote, if you have a field where are inside the myocardial fiber would be great.

Response: With the reduction of the pathological component and with the reformulation of the results. This figure was removed.

D. I strongly recommend taking a photomicrograph that expresses the necrotizing component of hepatitis, otherwise you should change the caption on this photo.

Response: With the reduction of the pathological component and with the reformulation of the results. This figure was removed.

E. In my opinion if you don't have a potential cause for this injury, I recommend removing it.

Response: With the reduction of the pathological component and with the reformulation of the results. This figure was removed.

F. I recommend removing this photo, since initially this is not a finding associated with an infectious component.

Response: With the reduction of the pathological component and with the reformulation of the results. This figure was removed.

Figure 2.

A. Surely you analyzed it, but to exemplify toxoplasmosis I think a microphotograph of a liver with tachyzoites and the lobular inflammatory and necrotizing process would be much more illustrative. If this was not possible, a pity, the photograph and the legend are fine, just a change, according to Dubey in the latest updates on toxoplasmosis, the term that should be used is no longer pseudocyst, now tissue cyst should be used.

Response: This change was accepted, figure 2 now becomes figure 3. The previous microphotograph was changed for a lung microphotograph where the inflammatory process and the presence of the tissue cyst are observed.

B. They should be proud of their histotechnician, those histological cuts are excellent!!!! I only recommend not mentioning the type of inflammatory cell since at this magnification it is not possible to identify them (even zooming in on the image).

Response: This change was accepted.

C. Excellent microphotography, the marvelous lesion!!! (poor primate), just a recommendation, we know that by culture you are sure of the bacteria, but in the histopathological slide my suggestion is to report with intralesional bacteria (Klebsiella pneumoniae by culture), if you want to place an insert I would suggest placing a photo of it histological gram field. Otherwise, I would remove the inset.

Response: This change was accepted. The inset was removed.

D. These lungworms were identified morphologically and/or by molecular techniques, or solely by the characteristics of the worm in the tissue. if it was solely for tissue characteristics please don't use gender, as we know that morphological keys are only suitable up to order or superfamily. It would be interesting to mention that in this cut they have both males and females (many more cuts of the female), since this helps to understand the cycle of this beautiful nematode. As in photo B, due to the magnification, it is not possible to identify the inflammatory cells.

Response: All nematodes were collected and identified by morphological characteristics until the gender level. This information appears in the current methodology on lines 154-158:

“All the parasites present in the carcasses were collected and washed with physiological saline, preserved in alcohol, acetic acid, and formalin (AFA) solution. No more than one week after collection they underwent identification to the genus level through morphometric characteristics. Physical and morphometric characteristics were recognized after fixation and clarification with Hoyer's solution by light microscopy. In addition, processed cestodes were stained with dilute Harris' hematoxylin solution.”

Likewise, this information is detailed in the legend of Figure 3, and the inset was removed.

E. Same previous comment on classification, additionally in that photo the nematodes are not illustrated (in the same way as in the other microphotographs).

Response: This microphotograph was removed

F. This microphotograph is not very illustrative, the different layers of the tissue are not observed, in my opinion this photo could be removed.

Response: This microphotograph was removed

G. Same previous comment on classification, additionally I want to tell you that this microphotograph is very pretty. I suggest that in the legend they mention the fibroplasia component (which is evident) and that it helps the morphological diagnosis of this Nodular and sclerosing gastritis condition.

Response: This change was accepted. The morphological diagnosis was corrected.

H. The photo is very good, but it is nothing more than a photo of a sarcocystis, if it were associated with an inflammatory process, I would recommend including it in the manuscript, otherwise I recommend removing it.

Response: This microphotograph was removed

I. Excellent microphotography, I don't know if this condition is reported in that particular species, congratulations.

Minor comments

Line 28: please add space in “the detection”.

Response: This change was made.

Line 30: please remove a space (there are double spaced) between words “a solid”

Response: This change was made.

Line 47-53: Dear author, I would like to better understand your statement, it represents a burden for society in what sense (economic due to the overload of health systems, due to treatment costs, etc.). In the manuscripts that you are citing there is interesting material for you to formulate a sentence with greater impact. I fully understand that the pandemic caused by SARS-CoV 2 can seduce us to cite it as an example; however there are other diseases that can serve as an example and can generate impacts for both humans and wildlife (e.g., yellow fever, tuberculosis, rabies, influenza, West Nile Virus (interesting example in your specific case :D) etc.). Respectfully, I encouraged you to write a new sentence.

Response: This paragraph was rephrased. The change appears on lines 48-53: 

“Zoonotic diseases represent a direct threat to public health systems, generating costs in terms of medical treatment, outbreak control and overloading health systems. In addition, it generates significant losses due to the slaughter of livestock and affectation of other domestic animals. Examples of how these diseases can impact public health, animal health and wildlife have been the recent outbreaks of yellow fever and West Nile virus, which show the need to have infrastructure and diagnostic capacity to ensure a constant surveillance of potentially zoonotic agents.”

Line 83-87: Dear authors, I considered that it would be interesting to add some additional information on the programs currently installed in some Latin American countries where wildlife is considered within the national surveillance programs (eg, non-human primates in yellow fever in Brazil [https://bvsms.saude.gov.br/bvs/publicacoes/guia_vigilancia_epizootias_primatas_entomologia.pdf]; in Argentina, game species are used to monitor some diseases such as trichinellosis [https://www.argentina.gob.ar/sites/default/files/informe_trichinellosis_2010_2019.pdf]; unfortunately those are the only two programs that I know of but with this information you could mention that there are some national programs installed and working perfectly where wild animals are used as sentinels for surveillance of some diseases).

Response: This change was accepted. The change appears on lines 80-82.

Line 139: change histological by histopathological.

Response: This change was accepted. The change appears on line 140.

Line 165: typing error “sender”

Response: This typing error was removed.

Line 166-167: I believe that in order to successfully replicate this study it would be interesting if the authors describe the criteria used to classify the age-range of the specimens.

Response: This information is included in the current methodology. The specimens were classified according to age, considering the development of the sexual organs and the phenotypic characteristics of the species. Because the classification by histological analysis of the teeth is really expensive. However, the juvenile specimens were very young, and it was easy to classify them.

Line 184-193: Does the number of 199 include all the organs evaluated in the pathology?

Response: Yes, all the organs evaluated in the pathology were included. Nevertheless, in the current version, this information was removed.

Final comments: Once again, I consider that this manuscript should be published since it brings a potential benefit to the scientific community that the readers of this prestigious journal will surely take advantage of. Obviously, the manuscript as it is at this moment needs some changes that I have no doubt that Dr. Alfaro-Alarcon and his team can make without any difficulty. I thank you for the opportunity to evaluate this important manuscript and I send you my best energies so that you can send it back with the suggested comments. Again, I remind you that I made this evaluation with great respect and my greatest critical sense so that you have a highly competitive manuscript.

---

## [Decision Letter · Decision Letter 1]

28 Jun 2022

PONE-D-21-39637R1Passive epidemiological surveillance in wildlife in Costa Rica identifies pathogens of zoonotic and conservation importancePLOS ONE

Dear Dr. Alfaro-Alarcon,

Thank you for submitting your manuscript to PLOS ONE. After careful consideration, we feel that it has merit but does not fully meet PLOS ONE’s publication criteria as it currently stands. Therefore, we invite you to submit a revised version of the manuscript that addresses the points raised during the review process.

Both reviewers agree that this version of the manuscript is improved, but the authors should strongly consider the points outlined below to make their suggestions on implementing a wildlife parasite surveillance system more generalizable. 

We look forward to receiving your revised manuscript.

Kind regards,

Daniel Becker

Academic Editor

PLOS ONE

Reviewers' comments:

Reviewer's Responses to Questions

**Comments to the Author**

1. If the authors have adequately addressed your comments raised in a previous round of review and you feel that this manuscript is now acceptable for publication, you may indicate that here to bypass the “Comments to the Author” section, enter your conflict of interest statement in the “Confidential to Editor” section, and submit your "Accept" recommendation.

Reviewer #1: (No Response)

Reviewer #2: All comments have been addressed

2. Is the manuscript technically sound, and do the data support the conclusions?

Reviewer #1: No

Reviewer #2: Yes

3. Has the statistical analysis been performed appropriately and rigorously? 

Reviewer #1: N/A

Reviewer #2: N/A

4. Have the authors made all data underlying the findings in their manuscript fully available?

Reviewer #1: Yes

Reviewer #2: Yes

5. Is the manuscript presented in an intelligible fashion and written in standard English?

Reviewer #1: No

Reviewer #2: Yes

6. Review Comments to the Author

Reviewer #1: The authors present a tool of passive surveillance that is both cost-effective and essential to monitor wildlife health and potential risk of spillover and spillback events. With their results, they demonstrate that despite the challenges developing countries face, there is a need to stay vigilant, as pathogens like Canine Distemper Virus and zoonotic gastrointestinal parasites were detected in their study. They also show and discuss important research gaps, such as the lack of information from certain areas in the country due to lack of resources and remoteness, and the misrepresentation of birds in their surveillance program.

In their first submission, the authors only presented their results without proposing a feasible and standardized method to implement this tool. In this version, they provide more detail (exemplified in figure 1), but I am afraid that their suggestions are very country specific. If this manuscript is accepted, I suggest that it would be under the study type “Methods, software, databases, and tools”, which should meet the criteria of utility, validation and availability (see https://journals.plos.org/plosone/s/submission-guidelines#loc-methods-software-databases-and-tools), and should be developed more broadly to propose a scheme for other countries with similar challenges.

Although, their epidemiological findings are interesting by themself, they are not the real focus of the paper, but rather are the evidence to stress the need to implement their proposed passive surveillance program in a sustainable manner with the tools that are already available. However, it seems like the authors shift the focus of their paper in presenting their findings. As an example, tables 3, 4 and 5 could be supplemental materials.

Finally, I found the manuscript difficult to follow. Occasionally, the sentences read awkward (e.g., Lines 50-53), had typos (e.g., Line 55) or wrong terms (e.g., “academy” in Lines 312-314), were grammatically incorrect (e.g., Line 80-82 and Line 95-97), or lacked a clear structure (e.g., Lines 108-112). The authors should do some intense editing to their manuscript, with a more concise language, and, if available, consider asking for English-proof reading services.

Reviewer #2: Dear authors, Initially, I want to express my good energies to you, for accepting the comments made and implementing them in an appropriate way in the manuscript. I confess that the manuscript with the modifications made is much clearer and meets the proposed objectives. A few comments below.

1. I recommend the use the following categories: metazoan parasites, protozoan parasites.

2. Please, double check the grammar of the text, mainly in the introduction, in general the English could be a little more fluent. However, I emphasize that this does not affect at all the understanding of the ideas they expose.

Minor comments:

Line 5: delete of (repeated)

Line 197: change pulmonary parasitic worms to lungworm.

Table 4: change parasitic worms to metazoan parasites.

For my part, I have no further comments, I believe that this manuscript will be of great importance to show feasible examples of disease surveillance in wildlife in Latin America. Once again congratulations Dr. Alfaro and coatures.

7. PLOS authors have the option to publish the peer review history of their article (what does this mean?). If published, this will include your full peer review and any attached files.

Reviewer #1: No

Reviewer #2: **Yes: **Pedro Enrique Navas-Suárez

---

## [Author Response · Author response to Decision Letter 1]

11 Aug 2022

Reviewer 1

In their first submission, the authors only presented their results without proposing a feasible and standardized method to implement this tool. In this version, they provide more detail (exemplified in figure 1), but I am afraid that their suggestions are very country specific. If this manuscript is accepted, I suggest that it would be under the study type "Methods, software, databases, and tools", which should meet the criteria of utility, validation and availability (see https://journals.plos.org/plosone/s/submission-guidelines#loc-methods-software-databases-and-tools), and should be developed more broadly to propose a scheme for other countries with similar challenges. Although, their epidemiological findings are interesting by themself, they are not the real focus of the paper, but rather are the evidence to stress the need to implement their proposed passive surveillance program in a sustainable manner with the tools that are already available. However, it seems like the authors shift the focus of their paper in presenting their findings. As an example, tables 3, 4 and 5 could be supplemental materials.

R/ We thank reviewer #1 for the constructive and helpful comments. The reviewers' suggestion to standardize a general passive wildlife surveillance scheme for countries with limited resources is an exciting objective. However, we consider that that idea exceeds the scope and intentions of this research. This manuscript aims to communicate our experience of what could be the establishment of a passive epidemiological surveillance system in the wildlife of Costa Rica as a model. This study is a pilot effort with the intention that the scientific community of other countries can visualize what aspects should be considered when aspiring for a surveillance system for wildlife animals. We believe that the current scope of this study can be helpful to countries pursuing this goal.

We also want the manuscript to illustrate aspects such as the type of agents detected and how to approach their diagnosis based on the available diagnostic capacities. This is essential in countries with low income dedicated to epidemiological surveillance of wildlife. In addition, we want to highlight the importance of linkage among human, animal, and wildlife health authorities and higher education institutions under the one health approach. This is based on the urgent need to raise awareness of the benefits of this type of disease monitoring for the countries and its regional and global impact. 

Based on this study's limitations, we have recognized the weaknesses in lines: #357-365. However, the proposed program is limited to Costa Rica's resources and infrastructure and does not apply to every country. Therefore, we emphasize that it is important to start evaluating the implementation of these programs in regions where disease surveillance in wildlife is minimal based on experiences like ours. Furthermore, although it is necessary to standardize methods and techniques for monitoring pathogens in wildlife, the development of pilot schemes allows sharing experiences with other programs already installed.

In tune with the European Wildlife Disease Association and OIE, although efforts have been made to expand and unify surveillance programs dedicated to wildlife, significant variations persist. There are still variations in the scope, scale, and capacity for establishing surveillance schemes in many regions. These limitations are mentioned in lines: #372-375. 

We are confident that in future studies, we will be able to compare our scheme with other WHMPs in the region to contribute to building these systems in other regions. In addition, other upcoming experiences would help standardize and optimize monitoring systems in the scientific community. Based on the reasons mentioned above and by recognizing the limitations of our study in the manuscript, we believe that the study should maintain the initially conceived structure and leave for future studies the pertinent approach proposed by reviewer 1.

Finally, I found the manuscript difficult to follow. Occasionally, the sentences read awkward (e.g., Lines 50-53), had typos (e.g., Line 55) or wrong terms (e.g., "academy" in Lines 312-314), were grammatically incorrect (e.g., Line 80-82 and Line 95-97), or lacked a clear structure (e.g., Lines 108-112). The authors should do some intense editing to their manuscript, with a more concise language, and, if available, consider asking for English-proof reading services.

R/We apologize for these mistakes. The English language and grammar were revised throughout the text, and the manuscript was improved to make it more concise and fluent.

Reviewer #2: 

We would like to thank reviewer #2 for the positive comments

I recommend the use the following categories: metazoan parasites, protozoan parasites.

This was modified as requested

Please, double check the grammar of the text, mainly in the introduction, in general, the English could be a little more fluent. However, I emphasize that this does not affect at all the understanding of the ideas they expose.

The English language was revised throughout the text.

Minor comments:

Line 5: delete of (repeated)

This was corrected

Line 197: change pulmonary parasitic worms to lungworm.

This was corrected

Table 4: change parasitic worms to metazoan parasites.

This was corrected

---

## [Decision Letter · Decision Letter 2]

30 Aug 2022

PONE-D-21-39637R2Passive epidemiological surveillance in wildlife in Costa Rica identifies pathogens of zoonotic and conservation importancePLOS ONE

Dear Dr. Alfaro-Alarcon,

Thank you for submitting your manuscript to PLOS ONE. After careful consideration, we feel that it has merit but does not fully meet PLOS ONE’s publication criteria as it currently stands. Therefore, we invite you to submit a revised version of the manuscript that addresses the points raised during the review process.

We look forward to receiving your revised manuscript.

Kind regards,

Daniel Becker

Academic Editor

PLOS ONE

Journal Requirements:

Reviewers' comments:

Reviewer's Responses to Questions

**Comments to the Author**

1. If the authors have adequately addressed your comments raised in a previous round of review and you feel that this manuscript is now acceptable for publication, you may indicate that here to bypass the “Comments to the Author” section, enter your conflict of interest statement in the “Confidential to Editor” section, and submit your "Accept" recommendation.

Reviewer #1: All comments have been addressed

2. Is the manuscript technically sound, and do the data support the conclusions?

Reviewer #1: Yes

3. Has the statistical analysis been performed appropriately and rigorously? 

Reviewer #1: N/A

4. Have the authors made all data underlying the findings in their manuscript fully available?

Reviewer #1: Yes

5. Is the manuscript presented in an intelligible fashion and written in standard English?

Reviewer #1: Yes

6. Review Comments to the Author

Reviewer #1: I want to congratulate the authors, not just for the relevance of their paper and the amount of work they did with the implementation of the passive surveillance, but also for the great improvement they made to present their findings, implications and recommendations. I just have a few very minor comments that will be easily corrected.

L34-36: “For instance, 60% (51/85) of the deaths were not associated with an infectious agent. In 67.4% (32/85) of the cases, deaths were not associated with an infectious agent, but an infectious agent was detected.”

These two sentences seem to be repeating the statement, and there seems to be a mistake: 32/85 is 37.6%, not 67.4%. Please check this. Same error in L191.

L37: The acronym “WHMP” is mentioned here, without first stating what does it stand for.

L84: Consider removing the acronym “LMIC” since it is not used in the rest of the paper, and it is confusing.

7. PLOS authors have the option to publish the peer review history of their article (what does this mean?). If published, this will include your full peer review and any attached files.

Reviewer #1: No

---

## [Author Response · Author response to Decision Letter 2]

5 Sep 2022

Departamento de Patología

 Escuela Medicina Veterinaria 

Universidad Nacional

Dr. Alejandro Alfaro, Ph.D.

alejandro.alfaro.alarcon@una.cr

Heredia, September 5, 2021

To

Academic Editor

Reviewers

PLOS ONE

Dear academic editor, dear reviewers,

We would like to thank you all for your observations which will improve our paper. We hope this article would be published soon, the published information could be of great value for wildlife scientists.

Journal Requirements

R/ References were checked:

References #23, #93 and #103 have been replaced with current and relevant references.

References #5, #7, #70 and #73 were removed as they were not relevant.

Reviewer 1

I want to congratulate the authors, not just for the relevance of their paper and the amount of work they did with the implementation of the passive surveillance, but also for the great improvement they made to present their findings, implications and recommendations. I just have a few very minor comments that will be easily corrected.

We would like to thank reviewer #1 for the positive comments.

L34-36: “For instance, 60% (51/85) of the deaths were not associated with an infectious agent. In 67.4% (32/85) of the cases, deaths were not associated with an infectious agent, but an infectious agent was detected.”

These two sentences seem to be repeating the statement, and there seems to be a mistake: 32/85 is 37.6%, not 67.4%. Please check this. Same error in L191.

This was corrected. Line 34-35: “For instance, 60% (51/85) of the deaths were not directly associated with an infectious agent. Though in 37.6% (32/85) of these cases an infectious agent associated or not with disease was detected.”

Line 191: the percentage was corrected.

L37: The acronym “WHMP” is mentioned here, without first stating what does it stand for.

This was corrected.

L84: Consider removing the acronym “LMIC” since it is not used in the rest of the paper, and it is confusing.

This was modified as requested.

Sincerely yours,

Dr. Alejandro Alfaro-Alarcón Ph.D.

Departamento de Patología

Escuela de Medicina Veterinaria

Universidad Nacional

alejandro.alfaro.alarcon@una.cr

---

## [Editor Report · Decision Letter 3]

11 Sep 2022

Passive epidemiological surveillance in wildlife in Costa Rica identifies pathogens of zoonotic and conservation importance

PONE-D-21-39637R3

Dear Dr. Alfaro-Alarcon,

We’re pleased to inform you that your manuscript has been judged scientifically suitable for publication and will be formally accepted for publication once it meets all outstanding technical requirements.

Kind regards,

Daniel Becker

Academic Editor

PLOS ONE
---

## [Editor Report · Acceptance letter]

14 Sep 2022

PONE-D-21-39637R3 

Passive epidemiological surveillance in wildlife in Costa Rica identifies pathogens of zoonotic and conservation importance 

Dear Dr. Alfaro-Alarcón:

I'm pleased to inform you that your manuscript has been deemed suitable for publication in PLOS ONE. Congratulations! Your manuscript is now with our production department. 

Kind regards, 

on behalf of

Dr. Daniel Becker 

Academic Editor

PLOS ONE